# Molecular mechanism of phosphopeptide neoantigen immunogenicity

Yury Patskovsky[1,2,4], Aswin Natarajan[1,2,4], Larysa Patskovska[1,2], Samantha Nyovanie[1,2], Bishnu Joshi[3], Benjamin Morin [3], Christine Brittsan[3], Olivia Huber[3], Samuel Gordon[3], Xavier Michelet[3], Florian Schmitzberger [3], Robert B. Stein[3], Mark A. Findeis [3], Andy Hurwitz[3], Marc Van Dijk [3], Eleni Chantzoura[3], Alvaro S. Yague[3], Daniel Pollack Smith[3], Jennifer S. Buell[3], Dennis Underwood [3] ✉ & Michelle Krogsgaard [1,2] ✉

Altered protein phosphorylation in cancer cells often leads to surface presentation of phosphopeptide neoantigens. However, their role in cancer immunogenicity remains unclear. Here we describe a mechanism by which an HLA-B*0702-specific acute myeloid leukemia phosphoneoantigen, pMLL$_{747-755}$ (EPR(pS)PSHSM), is recognized by a cognate T cell receptor named TCR27, a candidate for cancer immunotherapy. We show that the replacement of phosphoserine P$_4$ with serine or phosphomimetics does not affect pMHC conformation or peptide-MHC affinity but abrogates TCR27-dependent T cell activation and weakens binding between TCR27 and pMHC. Here we describe the crystal structures for TCR27 and cognate pMHC, map of the interface produced by nuclear magnetic resonance, and a ternary complex generated using information-driven protein docking. Our data show that non-covalent interactions between the epitope phosphate group and TCR27 are crucial for TCR specificity. This study supports development of new treatment options for cancer patients through target expansion and TCR optimization.

The magnitude of the T cell immune response to cancer typically correlates with neoantigen burden[1]. However, high tumor heterogeneity and the lack of shared neoepitopes between patients reduces the impact of neoantigen-based therapies as they are challenging to implement and difficult to generalize[2]. This problem prompted the search for alternative shared neoantigens that may originate from post-translational protein modifications (PTM). PTM that result in the altered function of protein kinases or phosphatases may increase protein phosphorylation that, in turn, can elevate lifetimes of intracellular phosphopeptides. As a result, these phosphopeptides can then appear on the surface of cancer cells as pMHC complexes[3-5]. The primary source of the dysregulation signature in tumor and virally infected cells is suppression of PP2A, a critical phosphatase that controls numerous signaling pathways[6]. Moreover, high levels of endogenous

PP2A inhibitors (SET proteins or CIP2A) are associated with cancer progression[7]. The decreased activity of protein phosphatase 1 was also associated with increased MHC presentation of phosphopeptides[8].

Although phosphopeptides have been detected in pMHC complexes isolated from both healthy and cancerous tissues, some phosphopeptide epitopes were cancer specific, most prominently in leukemias and melanoma[3,4,9-12]. In support of phosphopeptides as potential targets in cancer therapy, Lin and colleagues demonstrated that transgenic mice immunized with a tumor necrosis factor receptor associated protein 1 (TRAP1) HLA-A2-specific phosphopeptide had delayed tumor growth and prolonged survival[13]. Furthermore, clinical studies in melanoma patients demonstrated safety and immunogenicity of vaccines containing the cancer-associated HLA-A2 phosphopeptides pBCAR3$_{126-134}$ and pIRS2$_{1097-1105}$[14].

[1]Department of Pathology, New York University Grossman School of Medicine, New York, NY, USA. [2]Laura and Isaac Perlmutter Cancer Center at NYU Langone Health, New York, NY, USA. [3]Agenus, Lexington, MA, USA. [4]These authors contributed equally: Yury Patskovsky, Aswin Natarajan. ✉e-mail: dennis.underwood@agenusbio.com; Michelle.Krogsgaard@nyulangone.org

The mechanism of phosphopeptide immunogenicity remains unexplored, and no pMHC-TCR structures containing phosphopeptide antigens have been reported. Several structures have been published for phosphopeptide epitopes in complex with the class I pMHC HLA-A*0201[15–17]. Nearly all of these peptides have a phosphoserine in position $P_4$ and the majority have a basic arginine or lysine residue in $P_1$, reflective of the phosphorylation pattern of a 1,4-basophilic protein kinase motif (e.g., Akt) and proline directed kinases (e.g., MAP kinases and CDKs)[18]. Such features, in combination with the basic nature of the HLA-A*0201 peptide-binding site, often lead to strong electrostatic interactions between arginine (or lysine) residues and the peptide phosphate group. This results in high affinity binding between the phosphopeptide and HLA and reduced solvent exposure for the phosphate group[15–17]. However, when the residue $P_1$ is not basic, the phosphate group at P4 is solvent exposed, and the affinity between HLA-A*0201 and phosphopeptide is similar to that of the non-phosphoepitope[16]. In the crystal structures of phosphopeptides in complex with HLA-B*4001[19] and HLA-DR1[20] the phosphoserine residues adopted a non-anchor orientation, were solvent-exposed and thus more accessible for TCR recognition. In contrast to HLA-A2, the arginine at $P_1$ did not affect phosphoserine $P_4$ conformation in HLA-B*4001[19], which remained largely solvent exposed. However, the relationship between phosphate exposure and epitope immunogenicity was unclear, as antigen-specific TCRs were not identified in the above studies.

A lack of mechanistic data has hampered efforts to understand the role of phosphorylated peptides in immunity and the scope of their potential applications in cancer immunotherapy. To address this problem, we determined the molecular mechanism by which the HLA-B*0702-specific phosphopeptide epitope $pMLL_{747–755}$ (EPR(pS)PSHSM) is recognized by its cognate TCR, which we identified and named TCR27-LC13 (or TCR27). The $pMLL_{747–755}$ epitope was previously identified in acute myeloid leukemia (AML) cells but not in healthy tissues[10], and its presence in leukemia has been detected in several high throughput proteomics studies[21]. The wild-type (WT) $MLL_{747–755}$ (EPRSPSHSM) peptide is encoded by the human gene histone-lysine N-methyl transferase 2A (KMT2A), commonly known as MLL.

Here we describe the crystal structures for TCR27 and for HLA-B*0702 in complex with $pMLL_{747–755}$ or similar peptides. Using these structures, TROSY-NMR protein interface mapping, and information-driven Haddock docking, we have assembled a ternary TCR-pMHC complex, which reveals the interacting residues at its interface, and illustrates the key role of phosphoserine $P_4$ in complex stability and T cell activation. These data may be instructive for the development and optimization of immunotherapies for AML, and potentially other cancers, utilizing shared phosphopeptide neoantigens for vaccination and adoptive T cell transfer strategies.

## Results

### The phosphate group is essential for TCR27-mediated T cell activation

The cognate TCR for $pMLL_{747–755}$, named TCR27, was identified from CD8+ $pMLL_{747–755}$-activated T cells from a healthy donor, and it contains the TRAV27 and TRBV27 variable gene segments. Here we will focus on the mechanistic characterization of TCR27. To determine epitope specificity, CD8+ T cells from healthy donors were transduced with lentivirus expressing TCR27, cultured with HLA-B*0702 positive T2 cells pulsed with $pMLL_{747–755}$, and assessed for expression of activation markers CD25 and CD69 (Fig. 1a). TCR27$^+$ T cells displayed a dose-dependent increase in activation markers in the presence of $pMLL_{747–755}$-treated T2 cells, whereas no activation was observed in the presence of $MLL_{747–755}$ (Fig. 1b). Since phosphomimetics are resistant to phosphatases, we considered replacing the phosphoserine at $P_4$ with one of its mimetics for our mechanistic studies as well as for any potential future therapeutic applications. To determine the role of pSer-$P_4$ in T cell activation, we replaced it with acidic residues

aspartate or glutamate, leading to considerably diminished or no T cell activation, respectively (Fig. 1b). Similarly, replacement of phosphoserine with phosphonate analog (E7P) resulted in low T cell activation, though it was the highest activation level among all the phosphomimetics tested (Fig. 1b and Supplementary Fig. 1a). The diminished T cell activation in response to peptides with substitutions at $P_4$ was attributed to the reduced affinity between pMHC and TCR27 and not altered stability of the pMHC complex itself, as each peptide binds HLA-B*0702 with nearly the same affinity (Supplementary Fig. 1b, c).

To further investigate the role of individual epitope residues in TCR27 recognition, we performed a peptide scan, in which a set of peptides synthesized with all possible amino acid replacements at all positions (except for the anchor residues Pro-$P_2$ or Met-$P_9$) were assessed for T cell activation potential. Residues Pro-$P_5$, His-$P_7$, and pSer-$P_4$ were intolerant to substitutions (Fig. 1c), while the $P_1$ position was relatively conserved, with only glutamic acid or leucine allowed. This indicates that the $P_1$ residue side chain is located close to TCR27 at the pMHC-TCR interface. Arg-$P_3$ is an anchor residue but not conserved, and was replaced with alanine, lysine or cysteine without a substantial decrease in T cell activation. Anchor Ser-$P_6$ could be substituted with asparagine or cysteine, but not with other residues, which correlated with the size of the corresponding hydrophilic cavity in the pMHC structure (see below). The majority of Ser-$P_8$ substitutions did not affect T cell activation.

TSPRINT analysis[22] provides a unique opportunity to determine if additional HLA-B*0702-restricted phosphopeptides encoded in the human genome could be recognized by TCR27, which may be useful in mechanistic studies and potential clinical applications. In this analysis, we generated a 9-residue search template (E/L)PxSPxHx(A/L/V/I/M) to evaluate the possible 9-mer peptides that are present in the human proteome[23] and that match the structure activity relationship (SAR) requirements obtained from the scan data. Peptides identified in this manner represented a set of possible, but not confirmed TCR27-reactive epitopes. TSPRINT identified a total of 50 human proteome hits, including $MLL_{747–755}$ and $DOT1L_{998–1006}$. However, aside from $pMLL_{747–755}$, only $pDOT1L_{998–1006}$ activated the TCR27+ T cells when loaded onto HLA-B*0702+ T2 cells (Fig. 1d), while the 10 other highest scored phosphopeptides tested were completely inactive. The amino acid sequence identity between $pMLL_{747–755}$ and $pDOT1L_{998–1006}$ is 45%. Both epitopes, $pMLL_{747–755}$ and $pDOT1L_{998–1006}$, activated TCR27+ T cells to a similar degree, something that was not seen for their non-phosphorylated analogs (Fig. 1b, d). Similarly, the cytotoxic effect of TCR27$^+$ T cells was similar for both $pMLL_{747–755}$ and $pDOT1L_{998–1006}$ (Supplementary Fig. 1d). The effect was HLA-B*0702-specific (Supplementary Fig. 1e).

Therefore, pSer-$P_4$ is crucial for TCR27+ T cell activation, as phosphate removal or replacement with phosphomimetics eliminated or sharply reduced TCR27+ T cell activation. The scan data and the cross-reactivity between the two phosphoepitopes suggest a common pMHC-TCR27 recognition motif encompassing the residues $P_4$–$P_7$.

### Phosphoserine P4 is essential for TCR27-pMHC complex stability

To characterize the physical interactions between pMHC ligands and TCR27 we utilized biolayer interferometry (BLI), which showed that the binding affinity between pMHC and TCR27 is dependent on the nature of the amino acid residue at $P_4$ (Fig. 2a). The highest affinity was observed for pMHC in complex with phosphopeptides $pMLL_{747–755}$ or $pDOT1L_{998–1006}$, with $K_D$ values of 1.9 μM and 7.7 μM, respectively. These values fall within the range typical for TCRs[24]. By contrast, the pMHC complexes with WT peptides $MLL_{747–755}$ and $DOT1L_{998–1006}$ did not interact with TCR27 ($K_D > 2$ mM). Substitution of phosphate with phosphonate (E7P-$MLL_{747–755}$) or sulfate (OSE-$MLL_{747–755}$) sharply weakened the pMHC-TCR27 interaction ($K_D = 47$ μM and $K_D > 200$ μM, respectively). The binding data align well with the T cell activation data described above (Fig. 1b, d; Supplementary Fig. 1d).

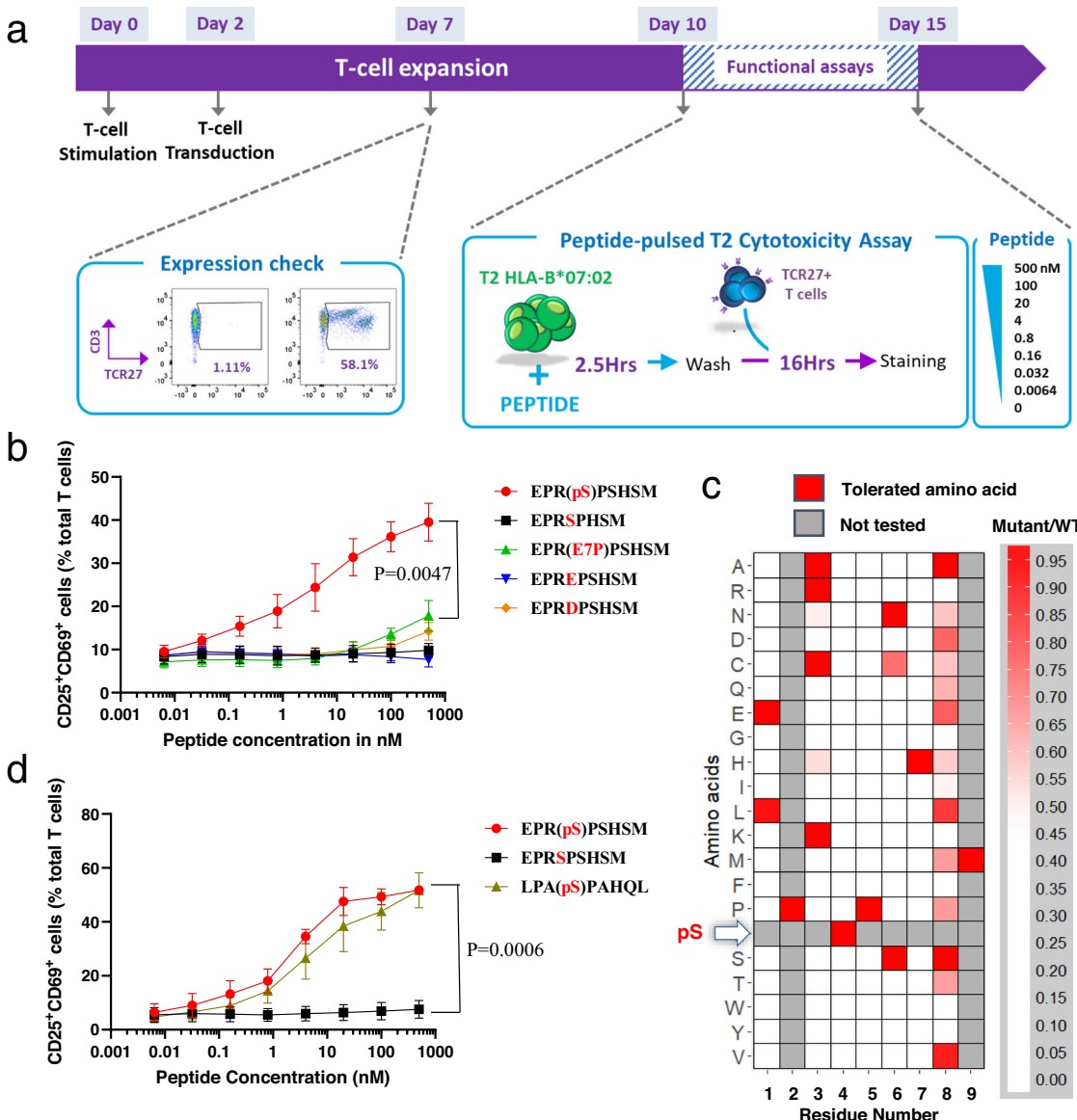

**Fig. 1 | Specificity of TCR27 towards pMLL$_{747-755}$ and pDOT1L$_{998-1006}$ epitopes.**
**a** Schematic representation of in vitro experimental procedures. T cells from healthy donors were isolated, activated and transduced with lentivirus expressing TCR27. TCR27 expression was analyzed at day 7 by flow cytometry. T cell cytotoxicity and activation were evaluated between days 10 and 15. H = hours. **b** TCR27-transduced T cell activation by different peptides. HLA-B*0702-expressing T2 (Target) cells were pulsed with various concentrations of pMLL$_{747-755}$ peptide and its analogs. Peptide-pulsed Target cells were co-cultured with TCR27-transduced T cells or with non-transduced (control) T cells. T cell upregulation of CD25 and CD69 was analyzed by flow cytometry in triplicate and expressed as mean ± SEM.

The shown in figure *P* value between the pMLL$_{747-755}$ and E7P- pMLL$_{747-755}$ (phosphonate) datasets was calculated using a two-sided Mann–Whitney *U* test.
**c** PMLL$_{747-755}$ and a set of the mutant peptides carrying single residue substitutions were analyzed in the T cell activation assay by detecting CD25/CD69 markers using flow cytometry. Peptide activity (as a mean value of triplicate) was color-coded according to the ratio of activation induced by mutant peptide to that induced by the WT peptide, in which the WT peptide activity is set at 1.0. **d** Comparison of epitope-specific T cell activation between pMLL$_{747-755}$ and pDOT1L$_{998-1006}$. Legends the same as in (**b**). The shown in figure *P* value between the pMLL$_{747-755}$ and MLL$_{747-755}$ datasets was calculated using a two-sided Mann–Whitney *U* test.

Substitution of pSer-P$_4$ in pMLL$_{747-755}$ with any other residue tested was deleterious for both T cell activation (see Fig. 1) and TCR-pMHC ternary complex stability (Fig. 2a), respectively. It is notable that among the phosphomimetics, the phosphonate analog E7P-MLL$_{747-755}$ induced the highest T cell activation and supported the highest affinity between pMHC and TCR27 with K$_D$ = 47 µM (as compared to K$_D$ = 1.9 µM for pMLL$_{747-755}$). Moreover, the phosphonate chemical structure is the most similar to that of phosphate, differing only by the replacement of the methylene group (E7P-MLL$_{747-755}$) with a bridging oxygen O$_γ$ (pMLL$_{747-755}$). Altogether, our data demonstrate that TCR27 functional avidity is largely dependent on the strength of association between pMHC and TCR,

with the integrity of the phosphate group essential for complex stability and T cell activation.

## Phosphoserine P$_4$ but not mimetics influences peptide-HLA interface

A 9-residue epitope will usually retain its overall conformation observed in pMHC following formation of the TCR-pMHC complex[25]. Therefore, comparison of pMHC structures with distinct bound peptides could reveal the possible effect of epitope AA sequence on TCR-pMHC interactions. To exclude the influence of different crystallization conditions on protein conformation, we generated fully isomorphous (having the same space group and very similar unit cell

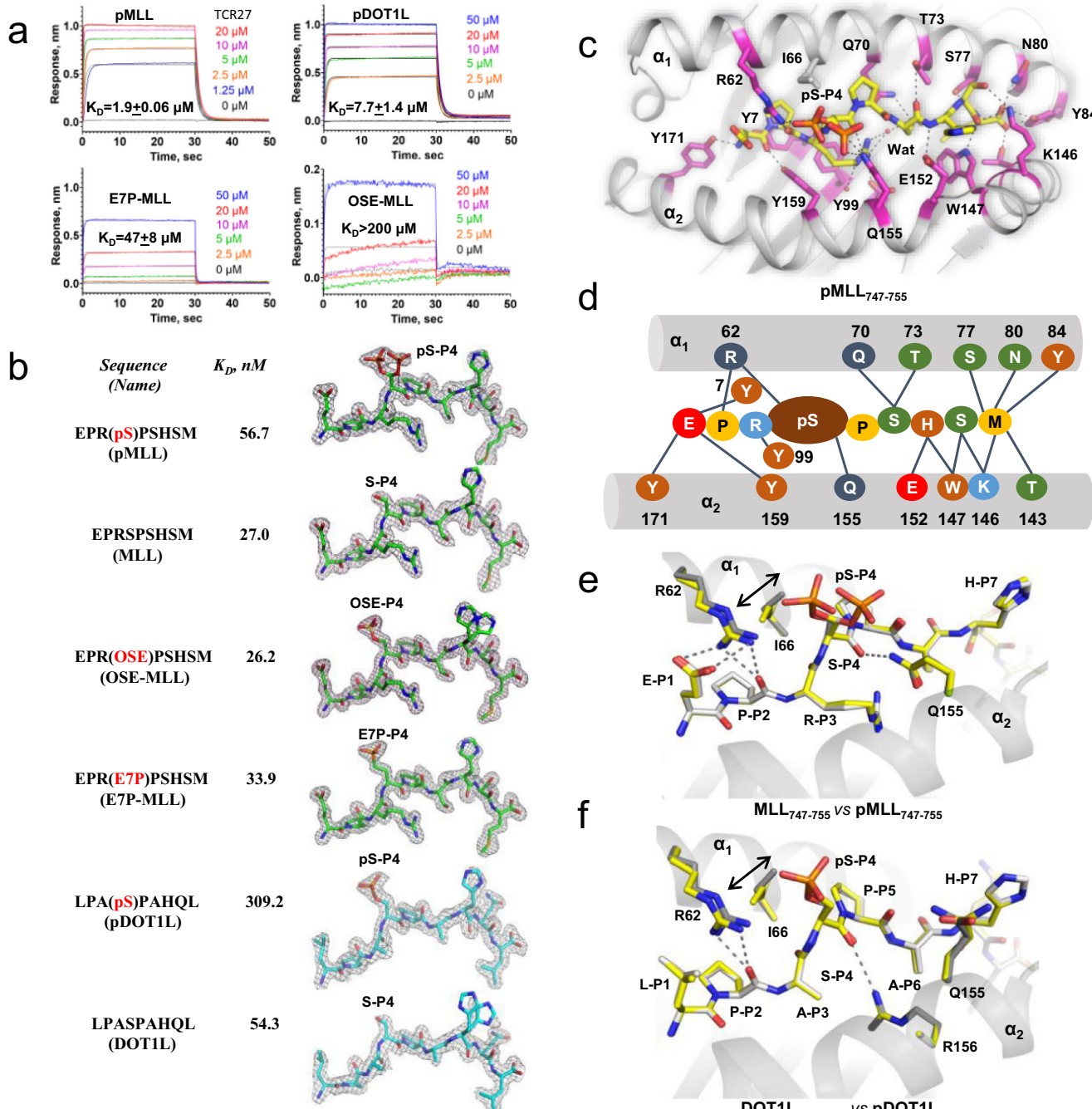

**Fig. 2 | High structural similarity between HLA-B*0702 complexes with different peptides. a** Sensorgrams demonstrate binding between TCR27 and pMHC, they were obtained by Bio-Layer Interferometry as described in Methods. $K_D$ values were determined using steady-state analysis. The difference between duplicates did not exceed 3%. **b** The structures of peptides in complex with HLA-B*0702. The sigmaA-weighted Fo-Fc (omit) electron density map (σ = 3.0, cutoff radius = 1.5 Å) was drawn around each peptide present in the corresponding pMHC crystal structure. Peptides $K_D$ values were determined using an Alpha assay as outlined in Methods. Residues others than Serine at $P_4$ are colored. **c** The inter-chain H-bond network inside the peptide-binding cavity of a binary pMLL$_{747-755}$/HLA-B*0702 complex (cartoon model). Individual amino acid residues are shown as sticks and the water molecules as non-bonded spheres. H-bonds presented as dotted lines.

**d** Scheme of the HLA-peptide interface for a binary pMLL$_{747-755}$/HLA-B*0702 complex. Amino acid residues were colored according to their properties (polar, nonpolar, basic, acidic, or aromatic); the hydrogen bonds are shown as lines and alpha helices as tubes. **e** Alignment of the crystal structures for pMLL$_{747-755}$/HLA-B*0702 (carbon atoms are gray) and MLL$_{747-755}$/HLA-B*0702 (carbon atoms are yellow). H-bonds (dotted lines) are shown between MLL$_{747-755}$ and Arg62. Individual residues are presented as sticks. The two alternate conformations for Ile66 are marked by a two-headed arrow. Hydrogen-bond distance cutoff was 3.5 Å. **f** Alignment of the crystal structures for pDOT1L$_{998-1006}$/HLA-B*0702 (carbon atoms are gray) and DOT1L$_{998-1006}$/HLA-B*0702 (carbon atoms are yellow). H-bonds are shown between DOT1L$_{998-1006}$ and Arg$_{62}$ or Arg$_{156}$.

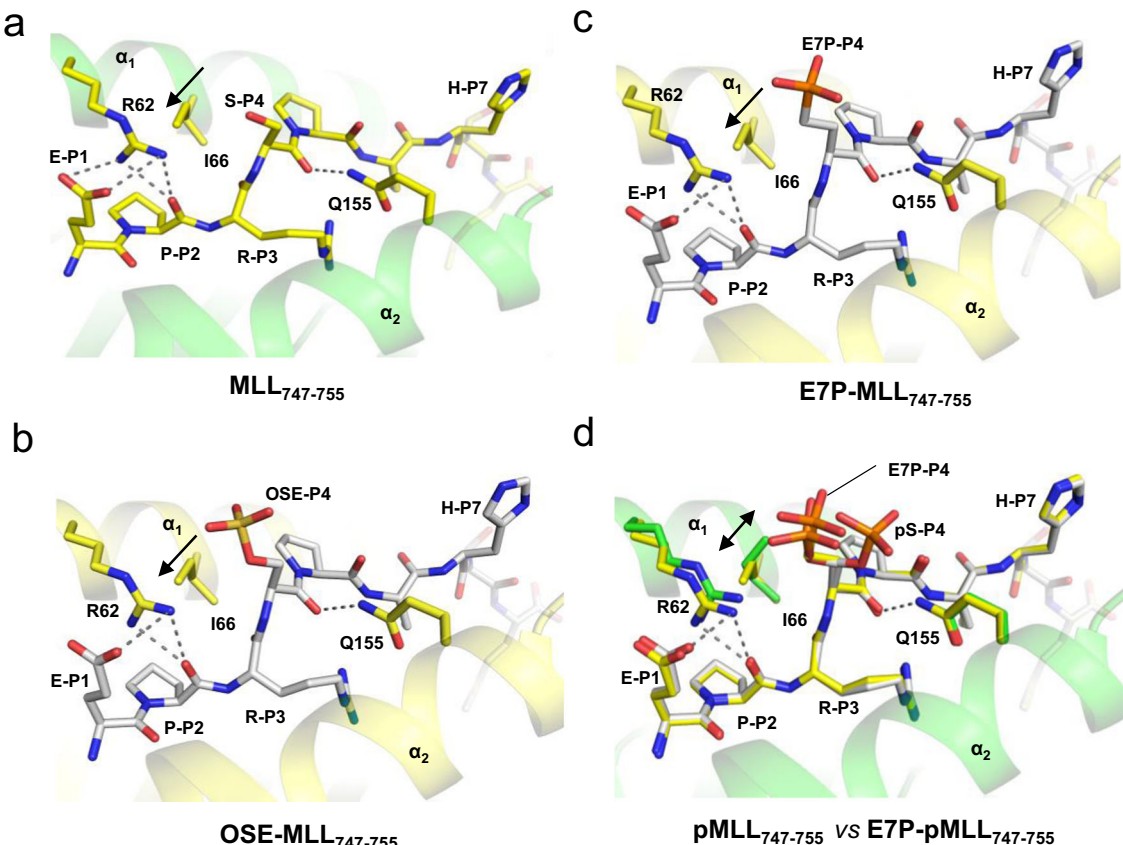

**Fig. 3 | The presence of phosphoserine at $P_4$ causes unique local conformational changes at the peptide-MHC binding site.** The amino acid residues are drawn as sticks. The Ile66 alternate conformations are marked by arrows. H-bonds (distance cutoff <3.5 Å) are the dotted lines between peptide and $Arg_{62}$ or $Gln_{155}$ residues, respectively. **a** Cartoon representation of the MLL$_{747-755}$/HLA-B*0702 structure, carbons are yellow. **b** Cartoon representation of the OSE-

MLL$_{747-755}$/HLA-B*0702 structure, peptide carbons are gray. **c** Cartoon representation of the E7P-MLL$_{747-755}$/HLA-B*0702 structure, peptide carbons are gray. **d** Superimposition of the E7P-MLL$_{747-755}$/HLA-B*0702 (carbon atoms are yellow) and pMLL$_{747-755}$/HLA-B*0702 crystal structures. H-bonds are shown between E7P-MLL$_{747-755}$ and $Arg_{62}$ or $Gln_{155}$.

parameters) complexes between HLA-B*0702 and the following peptides: MLL$_{747-755}$, pMLL$_{747-755}$, DOT1L$_{998-1006}$, pDOT1L$_{998-1006}$, E7P-MLL$_{747-755}$ and OSE-MLL$_{747-755}$ (Supplementary Table 1). Comparison of the two non-isomorphous pMLL$_{747-755}$/HLA-B*0702 structures (#1 and #2 in Supplementary Table 1) revealed conformational heterogeneity in the pMHC outer loops, mainly associated with different crystal packing, as seen in the corresponding CA-CA difference plot (Supplementary Fig. 2a). The pMLL$_{747-755}$ or MLL$_{747-755}$ conformations were also dependent on the presence of small molecules such as ethylene glycol or glycerol (cryoprotectants) that could bind inside the HLA pocket E (See Supplementary Fig. 5). To avoid that, we used sucrose for cryo-protection of crystals. In contrast, the isomorphous pMHC structures shared high similarity with core RMSD values of around or less than 0.1 Å between each other (Supplementary Fig. 2b), which allowed for the observation of small conformational differences associated with binding of distinct epitopes. Each solved pMHC complex had a typical MHC-I-fold, and every peptide amino acid sequence was validated during crystallographic refinement ($R_{free}$ value for the ligand was around or below 0.3), the quality of which is reflected by the near perfect match between the refined structures and the corresponding sigmaA-weighted 2Fo-Fc electron density maps (Fig. 2b). Our data indicate that every peptide binds in the same position between the HLA helices $α_1$ and $α_2$ (Fig. 2c). In each peptide, residue $P_4$ adopts a non-anchor orientation, which accounts for the minimal effect of amino acid substitutions at $P_4$ on the affinity between the corresponding peptides and HLA-B*0702 (Fig. 2b). The serine-phosphoserine replacement only slightly decreased both pMLL$_{747-755}$

and pDOT1L$_{998-1006}$ $K_D$ values, despite the salt bridge between the phosphate group of pSer-$P_4$ and the guanidine group of $Arg_{62}$. This bond likely does not contribute significantly to pMHC stability. In line with such a notion is the fact that pSer-$P_4$ in the pMLL$_{747-755}$/HLA-B*0702 structure adopted two alternate conformations—one with phosphate H-bonded to $Arg_{62}$, and a second one (with occupancy refined to 0.3) with phosphate H-bonded to the $Gln_{155}$ side chain (Fig. 2e). A strong salt bridge between phosphate and more basic $Arg_{62}$ would have prevented the second conformation. The H-bond network between pMLL$_{747-755}$ and HLA-B*0702 (Fig. 2c, d) reflects a typical HLA-epitope bonding pattern, wherein epitope's N- and C-termini and especially polar backbone groups are important contributors to the binary complex stability. Buried water molecules help to stabilize the Arg-$P_3$ and Ser-$P_6$ side chains at the binding site (Fig. 2c).

When compared to their corresponding WT peptides or phosphomimetics, the pMHC structures with phosphopeptides share high overall similarity between each other (Fig. 2e, f, Supplementary Fig. 3) except for minor conformational perturbations around $Arg_{62}$ and $Ile_{66}$. In structures with WT MLL$_{747-755}$ or DOT1L$_{998-1006}$ peptides, no protein atoms interacted with the hydroxyl groups of Ser-$P_4$. However, in the structures with phosphoepitopes, the guanidine group of $Arg_{62}$ was H-bonded to the phosphate moiety (Figs. 2e, f). Formation of this bond required a 0.5 Å shift of the $Arg_{62}$ side chain towards phosphate, which occurs in concert with $Ile_{66}$ switching to another alternate conformation (as indicated by the arrows in Fig. 2e, f and in Fig. 3) to prevent interatomic clashes between $Arg_{62}$ and $Ile_{66}$. In contrast, neither phosphonate nor sulfate maintain contacts with HLA atoms in the

corresponding pMHC structures, they are fully exposed to solvent and surrounded by water (Fig. 3a–c and Supplementary Fig. 4). Moreover, the structures with E7P-MLL$_{747–755}$ and OSE-MLL$_{747–755}$ peptides were nearly identical to that with the wild-type peptide MLL$_{747–755}$, including exactly the same positions for the Arg$_{62}$ and Ile$_{66}$ side chains (Fig. 3, Supplementary Fig. 2b). Therefore, the presence of phosphoserine but not of any other studied residue at P$_4$ caused unique structural perturbations at the HLA-epitope interface. It is evident that the slight differences between the chemical structures of phosphate and phosphomimetics were sufficient to induce such perturbations. For instance, in the pMLL$_{747–755}$/HLA-B*0702 complex, the O$_\gamma$ atom of pSer-P$_4$ is located within hydrogen-bond distance (<3.5 Å) from the Arg$_{62}$ guanidine group. However, in the structure with phosphonate (E7P-MLL$_{747–755}$) the same atomic position is occupied by the nonpolar methylene group not supporting hydrogen bonds, and thus the guanidine group of Arg$_{62}$ remains about 4 Å away from the C$_\gamma$ atom of the E7P residue. In conclusion, replacement of serine P$_4$ with phosphoserine but not with phosphomimetics leads to unique conformational changes at the ligand-HLA interface. As shown below, Arg$_{62}$ is also a part of the pMHC-TCR27 interface, and its orientation could influence interactions between Arg$_{62}$ and pSer-P$_4$ or TCR27, which are important for ternary complex stability.

## Basis for recognition between TCR27 and two distinct phosphopeptides

We compared the pMHC crystal structures of HLA-B*0702 in complex with pDOT1L$_{998–1006}$, pMLL$_{747–755}$ and their corresponding non-phosphopeptides to determine a common recognition motif for TCR27. Both crystal structures were isomorphous and shared a similar fold (core RMSD value between their C$_\alpha$ atoms was 0.22 Å) while displaying some local conformational differences. For instance, binding between pDOT1L$_{998–1006}$ and HLA-B*0702 resulted in the N-terminus of the α$_2$ helix located approximately 0.5 Å away from the α$_1$ helix, as compared to the structure with pMLL$_{747–755}$ epitope. In addition, the C-terminal segment of pDOT1L$_{998–1006}$ was shifted along with the α$_2$ helix (Fig. 4a and Supplementary Fig. 3b). By contrast, the N-terminus of pDOT1L$_{998–1006}$, including pSer-P$_4$, adopts a conformation very similar to that in pMLL$_{747–755}$. The helical "shift" was not driven by the phosphorylation status of the epitope, but it was a result of amino acid discrepancies in positions P$_3$ and P$_6$, and the magnitude of the observed shifts was approximately the same with either pDOT1L$_{998–1006}$ or DOT1L$_{998–1006}$ (Supplementary Fig. 3b). In the pMLL$_{747–755}$/HLA-B*0702 structure, the guanidine group of Arg-P$_3$ is H-bonded to the Asp$_{114}$ and Tyr$_{116}$ (both in strand E) side chain atoms, whereas the Arg$_{156}$ (α$_2$ helix) basic side chain is H-bonded to the Glu$_{152}$ and Gln$_{155}$ (both in α$_2$ helix) side chains. The substitution of Arg-P$_3$ and Ser-P$_6$ (pMLL$_{747–755}$) with alanine residues (pDOT1L$_{998–1006}$) leads to the distinct hydrogen bonding pattern with pDOT1L$_{998–1006}$ epitope, where Arg$_{156}$ is connected to the buried Asp$_{114}$ carboxyl group and the pSer-P$_4$ backbone oxygen instead (Fig. 4b). In the pMHC structures with DOT1L$_{998–1006}$ and pDOT1L$_{998–1006}$, the N-termini of α$_2$ helices are positioned farther away from the α$_1$ helices to accommodate the bulky Arg$_{156}$ side chain inside the peptide-binding cavity and to avoid clashes between the neighboring atoms (see Supplementary Fig. 3b). The helical shift affects the Gln$_{155}$ side chain orientation, which may explain the lack of alternate epitope conformation #2 with pDOT1L$_{998–1006}$ that is supported by the H-bonds between Gln$_{155}$ and pSer-P$_4$ of bound pMLL$_{747–755}$ in the corresponding pMHC structure (Fig. 4b). Based on the similarities and differences between the crystal structures, we conclude that the common pMHC recognition motif for TCR27 includes the epitope residues from pSer-P$_4$ to His-P$_7$ and adjacent HLA residues, but not the N-terminal segment of the HLA α$_2$ helix and the penultimate C-terminal residues for both, pDOT1L$_{998–1006}$ and pMLL$_{747–755}$. The proposed recognition pattern is supported by the level of solvent exposure for epitope residues. As a rule, exposed epitope residues are involved in

TCR recognition[25]. We have compared the solvent accessible surface area (SASA) for epitope residues in the MHC structures with both phosphopeptides (Fig. 4c–f). It is notable that pSer-P$_4$ remains solvent accessible in both structures, as is His-P$_7$ and, to lesser extent, Pro-P$_5$. These residues were invariable in the peptide scan. Gln-P$_8$ in pDOT1L$_{998–1006}$ is solvent exposed, but the pMLL$_{747–755}$ epitope has serine in this position, which was variable in the peptide scan, and thus P$_8$ was deemed as nonessential for TCR binding.

In summary, our data confirms the existence of a shared recognition motif that includes the pMLL$_{747–755}$ and pDOT1L$_{998–1006}$ invariant amino acid residues P$_4$, P$_5$, and P$_7$. However, despite cross-reactivity to TCR27 and high conformational similarity, the structures with pMLL$_{747–755}$ and pDOT1L$_{998–1006}$ were not identical, which is expected given that the affinity between TCR27 and pMHC was about 4-fold greater with pMLL$_{747–755}$ epitope (see Fig. 2a). In contrast to pMLL$_{747–755}$, the pMHC structure with DOT1L$_{998–1006}$ shows no alternate conformations for pSer-P$_4$. This observation allowed us to select the correct conformation of pSer-P$_4$ (H-bonded to Arg$_{62}$) for TCR-pMHC docking.

## TCR27 heterodimer displays conformational heterogeneity

The crystal structure of TCR27 ectodomain was determined by the X-ray crystallography. To solve the crystal structure of TCR27 by molecular replacement, we used the coordinates of a TCR that recognizes the MP$_{58–66}$ epitope (PDB 1OGA) with the highest amino acid sequence identity to TCR27 (93% between the TCRα chains). As a result, we obtained two types of non-isomorphous TCR27 crystals using different sets of crystallization conditions, and the corresponding structures were named TCR_1 (space group P2$_1$) and TCR_2 (space group P2$_1$2$_1$2$_1$) (Supplementary Table 1, Fig. 5a–c). The overall RMSD value between TCR27 (TCR_1) and TCR 1OGA was 1.4 Å and TCR27 adopted a typical TCR fold (Fig. 5b)[25]. The quaternary conformation of TCR_1 was different from that of TCR_2, with an RMSD value between protein atoms of the two structures of 3.1Å (with both CDR3 loops excluded from alignment), which indicated the existence of conformational heterogeneity in TCR27. The structural alignment between TCR_1 and TCR_2 is presented in Fig. 5c, in which we aligned the atomic coordinates of the Vα domains to illustrate the difference between the relative orientations of the distinct TCR subunits in each structure. To determine the mechanism of quaternary transition between TCR_1 and TCR_2, we compared the conformations of individual domains of the two TCRs. The residue-by-residue CA difference plots between the Vα and Vβ aligned protein domains demonstrated that the tertiary structures for both domains were similar in TCR_1 and TCR_2 despite a significant difference between their quaternary conformations (Fig. 5d–g). It is evident that the quaternary "transition" between TCR_1 and TCR_2 is mainly the result of the rigid-body rotation of one TCR subunit with respect to another one. The AA residues with the smallest differences between CA atoms, marked by arrows in Fig. 5d and f, indicate the position of the relative center of such rotation, or rotation axis.

This result does not rule out the possibility that both TCR_1 and TCR_2 structures can exist in close to physiological conditions, and their existence reflects conformational flexibility in TCR27. To date, however, there are no examples of two highly divergent quaternary conformations for the same TCR, and this problem requires further study. In our study, the conformational heterogeneity in TCR27 could have impaired our efforts to obtain the crystals of a ternary complex, the model of which was instead generated using the solved crystal structures of its components (pMHC and TCR), and the combination of the NMR-based interface mapping and information-driven molecular docking.

## Crystal structure of pMHC represents its conformation in solution

To aid information-driven docking and for generation of a ternary complex between pMLL$_{747–755}$/HLA-B*0702 and TCR27, a TROSY-NMR

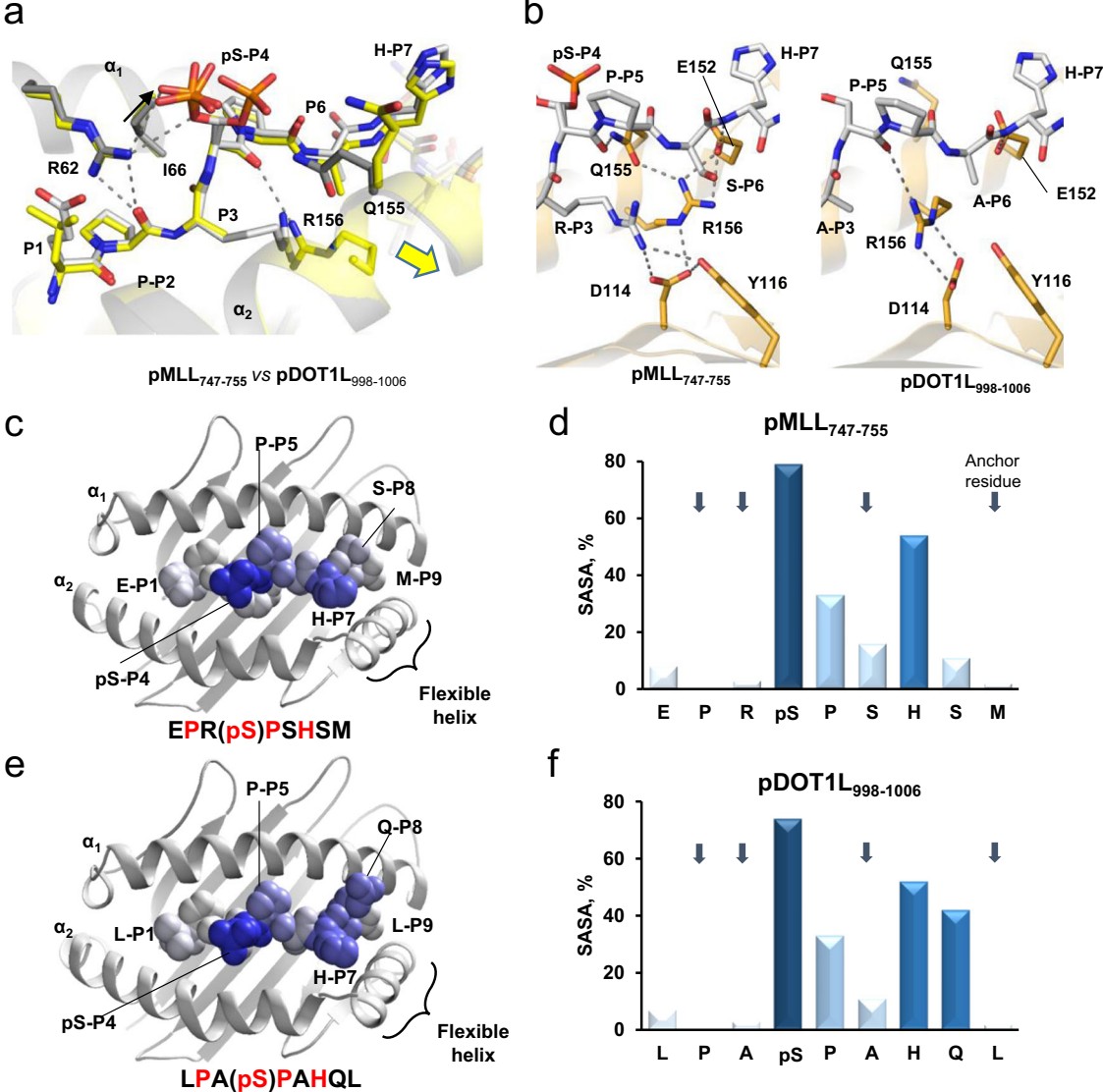

**Fig. 4 | Similarity between the pMHC structures with pMLL$_{747-755}$ and pDOT1L$_{998-1006}$ epitopes indicates the presence of a shared TCR recognition motif.** In **a** and **b**, amino acid residues are sticks, H-bonds (distance cutoff <3.5 Å) - dotted lines. **a** Alignment of the crystal structures for pMLL$_{747-755}$/HLA-B*0702 (carbon atoms are gray) and DOT1L$_{998-1006}$/HLA-B*0702 (carbon atoms are yellow). H-bonds are shown between DOT1L$_{998-1006}$ and Arg$_{62}$ or Arg$_{156}$. The arrow points to the direction of a helical shift. **b** Distinct H-bond patterns (bonds are dotted lines) in the crystal structures of pMLL$_{747-755}$/HLA-B*0702 and pDOT1L$_{998-1006}$/HLA-B*0702. Only HLA residues with different conformations and/or interaction patterns are shown. The carbon atoms in epitopes are colored in gray. Individual residues in pMLL$_{747-755}$ (**c**) or pDOT1L$_{998-1006}$ (**e**) peptides (all atoms are presented as Van-der-Waals spheres) are shaded according to relative solvent exposure (from none - white to 100% - dark blue). The cartoon models display the respective peptide-binding sites in HLA-B*0702. The minor alternate conformation of pSer-P$_4$ in the pMHC structure with pMLL$_{747-755}$ was omitted for clarity. Identical residues in epitope sequences are red-colored. Relative solvent-exposed surface area (SASA) is presented as a percentage of all surface area (ASA) for each epitope residue in the crystal structures of pMLL$_{747-755}$/HLA-B*0702 (**d**) or pDOT1L$_{998-1006}$/HLA-B*0702/(**f**). Anchor residues are marked by down arrows.

method was used to map pMHC residues located at the pMHC-TCR interface. First, we assigned NMR chemical shifts for the vast majority of HLA-B*0702 residues, including those located at the HLA-peptide interface, involved in epitope recognition and binding to TCR (Supplementary Fig. 6). To determine the effect of serine-phosphoserine substitution on the NMR spectra, we analyzed the differences in chemical shifts between the pMLL$_{747-755}$/HLA-B*0702 and MLL$_{747-755}$/HLA-B*0702 complexes by calculating the scaled chemical shift perturbations (CSP) in the $^{15}$N-TROSY spectra (Fig. 6a, b, Supplementary Table 2). The highest CSP values were observed for the $\alpha_1$ helical residues – Glu$_{58}$, Tyr$_{59}$, Asp$_{61}$, Ile$_{66}$, Tyr$_{67}$, Lys$_{68}$, Ala$_{69}$, Gln$_{70}$, Thr$_{73}$, Arg$_{75}$, and Ser$_{77}$, and for the $\alpha_2$ helical residues – Trp$_{147}$, Arg$_{151}$, Glu$_{152}$, Ala$_{153}$, Glu$_{154}$, Gln$_{155}$, Arg$_{156}$, Trp$_{167}$, Tyr$_{171}$, Leu$_{172}$, and Glu$_{173}$. The fact that CSPs are seen in residues that are distant from pSer-P$_4$ suggests

subtle differences between the pMHC complexes, including some that may be too small to be observed in the crystal structures. In general, however, there was a good agreement between the NMR and crystallographic data. For instance, the highest CSP values ($\delta > 0.1_{ppm}$) were detected among the HLA-B*0702 residues surrounding the phosphate moiety, such as Ile$_{66}$, Gln$_{70}$, Glu$_{152}$, Ala$_{153}$ and Gln$_{155}$. Arg$_{62}$ was excluded from CSP calculations due to spectral overlaps. The amino acid residues with the highest CSP values could be grouped into 4 separate clusters (Fig. 6a). The location of each cluster in the crystal structure is shown in Fig. 6c. Two adjacent amino acid clusters (#1 and #2) are located on the $\alpha_1$ helix, while two others (#3 and #4) are located on the N- and C-terminus of the $\alpha_2$ helix. Cluster #1 incorporates the residues neighboring Arg$_{62}$, the guanidine group of which is hydrogen bonded to phosphate in the crystal structure (Fig. 6d). Cluster #2 includes Ile$_{66}$

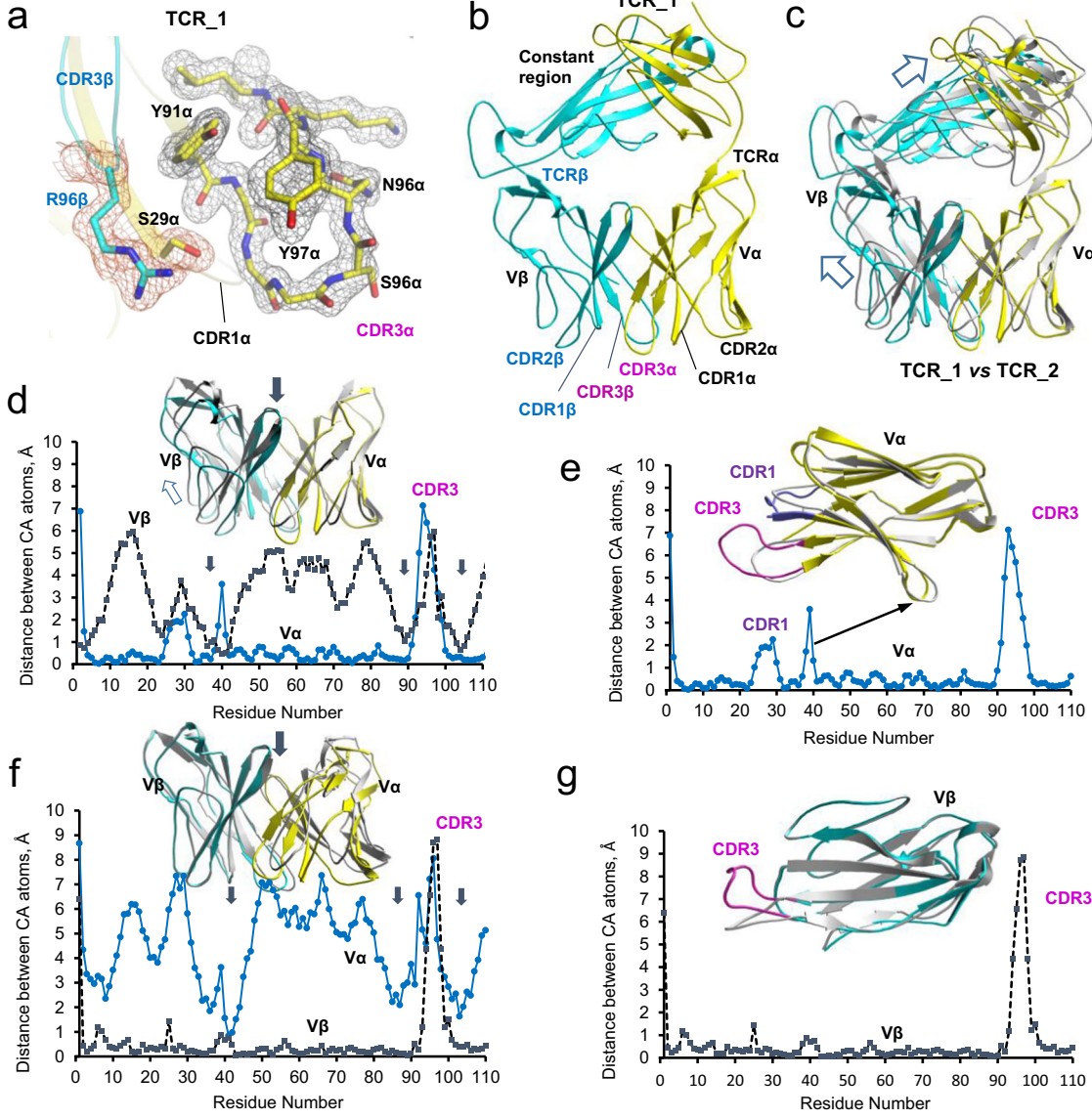

**Fig. 5 | The crystal structure of TCR27 is similar to other human TCRs but displays large conformational heterogeneity.** Cartoon representation. **a** TCR_1 structure. The sigmaA-weighted 2Fo-Fc electron density map is shown around amino acid residues that belong to the CDR3 loops. Individual residues are stick models. **b** Overall structure of the TCR27 (TCR_1) heterodimer, including locations of the separate chains, constant or variable regions, and the CDR loops. **c** Superposition of the TCR_1 and TCR_2 structures was performed using the atomic coordinates of the corresponding Vα domains. **d**, **e** Structural alignment of the TCR_1 and TCR_2 variable regions using the coordinates of the corresponding Vα domains. The Cα difference plots between TCR_1 and TCR_2 variable regions (**d**) or

the isolated Vα domains only (**e**) show similarity between their tertiary structures except for different orientations of the CDR3α loops. The transition between the two quaternary conformations is a result of the Vβ domain rotation around the center of rotation, an approximate position of which is indicated by the down arrows. **f**, **g**. Structural alignment of TCR_1 and TCR_2 variable regions using the coordinates of the corresponding Vβ domains. The Cα difference plots between the TCR_1 and TCR_2 variable regions (**f**) or the isolated Vβ domains only (**g**) demonstrate increased flexibility of the CDR3β loops. The transition between the two conformations could be the result of Vα domain "rotation" around the rotation axis, the approximate position of which is depicted by the down arrows.

and surrounding residues, chemical shifts of which could be directly or indirectly influenced by Ile$_{66}$. Cluster #3 includes residues close to Gln$_{155}$, the side chain of which in the crystal structure interacts with phosphate in the alternate conformation and is hydrogen bonded to the backbone oxygen atom of pSer-P$_4$. Finally, cluster #4 encompasses the residues close to Glu-P$_1$, the carboxyl group of which is within the H-bond distance from the guanidine group of Arg$_{62}$ in the X-ray structure. CSP clustering reflects the high sensitivity of TROSY-NMR in detecting small perturbations in residues surrounding the ligand-binding site[26,27]. Overall, the TROSY-NMR CSP data indicate that the presence of pSer-P$_4$ induces local structural perturbations around the same HLA-B*0702 amino acid residues that were identified in the crystal structure. The highest CSP values were observed for residues

located predominantly in the vicinity of or in direct contact with pSer-P$_4$. The NMR study demonstrated high homology between the solution and crystal structures and validated the NMR-based approach that was further utilized in mapping the TCR-pMHC interface.

## TCR-pMHC interface residues were mapped by TROSY-NMR

To identify the HLA residues that participate in the pMHC-TCR interface, we performed TROSY-NMR titration experiments by increasing the ratios between perdeuterated TCR27 and perdeuterated $^{15}$N-labeled HLA-B*0702 in complex with pMLL$_{747-755}$. Upon assembly, protein residues at the protein-protein interface experience positional changes in the chemical shifts (CSP), mostly because of a changing environment. In addition, strong protein-protein interactions usually

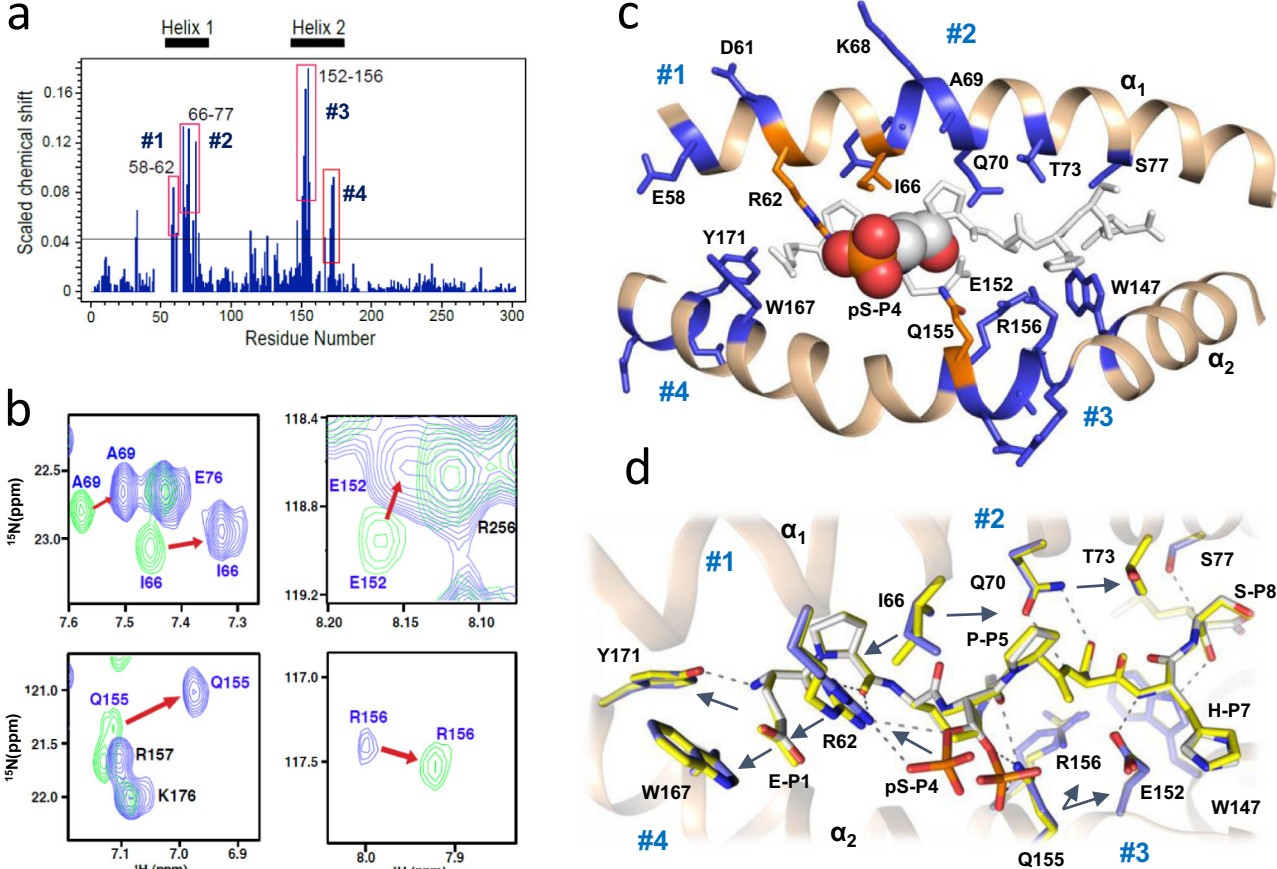

**Fig. 6 | NMR mapping of the HLA-epitope interface reveals consistency between the pMHC crystal structure and its conformation in solution. a** Plot of scaled chemical shift changes (CSP) versus residue number between $^{15}$N-TROSY spectra of pMLL$_{747-755}$/HLA-B*0702 and MLL$_{747-755}$/HLA-B*0702 complexes. The line at CSP = $0.043_{ppm}$ indicates a threshold at 1δ over mean CSP. The four amino acid clusters, #1 (58–62), #2 (66-77), #3 (152–156) and #4 (167–173) with highest CSP values are outlined. **b** Zoomed-in view of the overlay of $^{15}$N-TROSY spectra of pMLL$_{747-755}$/HLA-B*0702 (green) and MLL$_{747-755}$/HLA-B*0702 complexes (blue) showing CSPs of residues (indicated by the arrows) belonging to clusters #2 and #3. **c** Cartoon view of the two alpha helices involved in epitope binding: NMR cluster

#1 surrounding Arg$_{62}$ (orange), NMR cluster #2 surrounding Ile$_{66}$ (orange), NMR cluster #3 around Gln$_{155}$ (orange), and NMR cluster #4 near Glu-P$_1$ (blue). The peptide is shown as gray sticks, and pSer-P$_4$ is presented as a space-filling model. The minor alternate conformation for pSer-P$_4$ was omitted for clarity. **d** Overlay of pMHC structures with pMLL$_{747-755}$ (carbons are gray) and MLL$_{747-755}$ (carbons are yellow). The carbon atoms in HLA-B*0702 are colored in blue (with pMLL$_{747-755}$) or yellow (with MLL$_{747-755}$), respectively. The dashed lines show H-bonds between pMLL$_{747-755}$ and HLA residues. The arrowheads indicate the AA residues, NMR spectra of which can be affected by neighbors, explaining the reason for peaks clustering observed in (**a**).

result in reduced NMR peak intensities if an association-dissociation event occurs at the intermediate or slow NMR relaxation time scale[28]. Therefore, by monitoring both parameters, CSPs, and peak intensity changes upon TCR-pMHC complexation (Supplementary Fig. 7), the protein residues involved in the interactions can be identified. When perdeuterated $^{15}$N-labeled HLA-B*0702/ MLL$_{747-755}$ complex was used as a negative control, no significant interactions with TCR27 were detected, as no changes were observed in CSPs or peak intensities in comparison with the NMR spectrum of the binary complex alone (Supplementary Fig. 7b). By contrast, binding between TCR27 and pMLL$_{747-755}$/HLA-B*0702 was accompanied by an overall reduction in peak intensities in the corresponding $^{15}$N-TROSY spectra (Supplementary Fig. 7c, average V/V$_0$ = 0.39, pMLL$_{747-755}$/HLA-B*0702: TCR27 (1:1)), indicating an increase in molecular tumbling time, which is evidence of a strong binding event. The localized CSPs and peak intensity changes were assigned to the individual HLA residues, as presented in Fig. 7a–c. The peptide-surrounding residues (indicated by the red arrows in Fig. 7a and c), such as Gln$_{70}$ and Thr$_{73}$ from the α$_1$ helix and Ala$_{149}$, Arg$_{156}$, Ala$_{158}$ from the α$_2$ helix, demonstrated increased CSP values (δ$_{ppm}$ > 0.016) upon complexation. Significant peak intensity changes (V/V$_0$ < 0.1) were detected for residues Asp$_{61}$, Ile$_{66}$, Tyr$_{67}$, Lys$_{68}$, Ala$_{69}$, Gln$_{70}$, Thr$_{73}$ and Glu$_{76}$ from the α$_1$ helix and Trp$_{147}$, Arg$_{156}$

and Glu$_{163}$ from the α$_2$ helix (Fig. 7b). The distribution of the NMR-mapped residues is shown schematically using the crystal structure for the pMLL$_{747-755}$ /HLA-B*0702 complex in Fig. 7d. Thus, the TROSY-NMR-based CSP-intensity analysis confirmed direct binding between TCR27 and HLA-B*0702 in complex with pMLL$_{747-755}$, and identified a number of HLA residues located at or close to the TCR-pMHC interface.

## TCR27-pMHC complex and mechanism of phosphopeptide immunogenicity

To understand the mechanism of phosphopeptide recognition by TCR27, we generated a ternary pMHC-TCR complex using an information-driven Haddock docking algorithm[29]. The crystal structures of TCR27 and the corresponding pMHC complexes with phosphopeptides served as starting models. Data obtained by NMR, peptide scan, mutagenesis and X-ray crystallography were utilized to define the probable interface residues and limit the docking bias. Haddock protein-protein data driven docking software was chosen based on previous reports that Haddock docking efficiency equals or exceeds that of other protein docking platforms[30], and because it applies an advanced docking algorithm to improve docking efficiency by using information about protein interfaces obtained by other

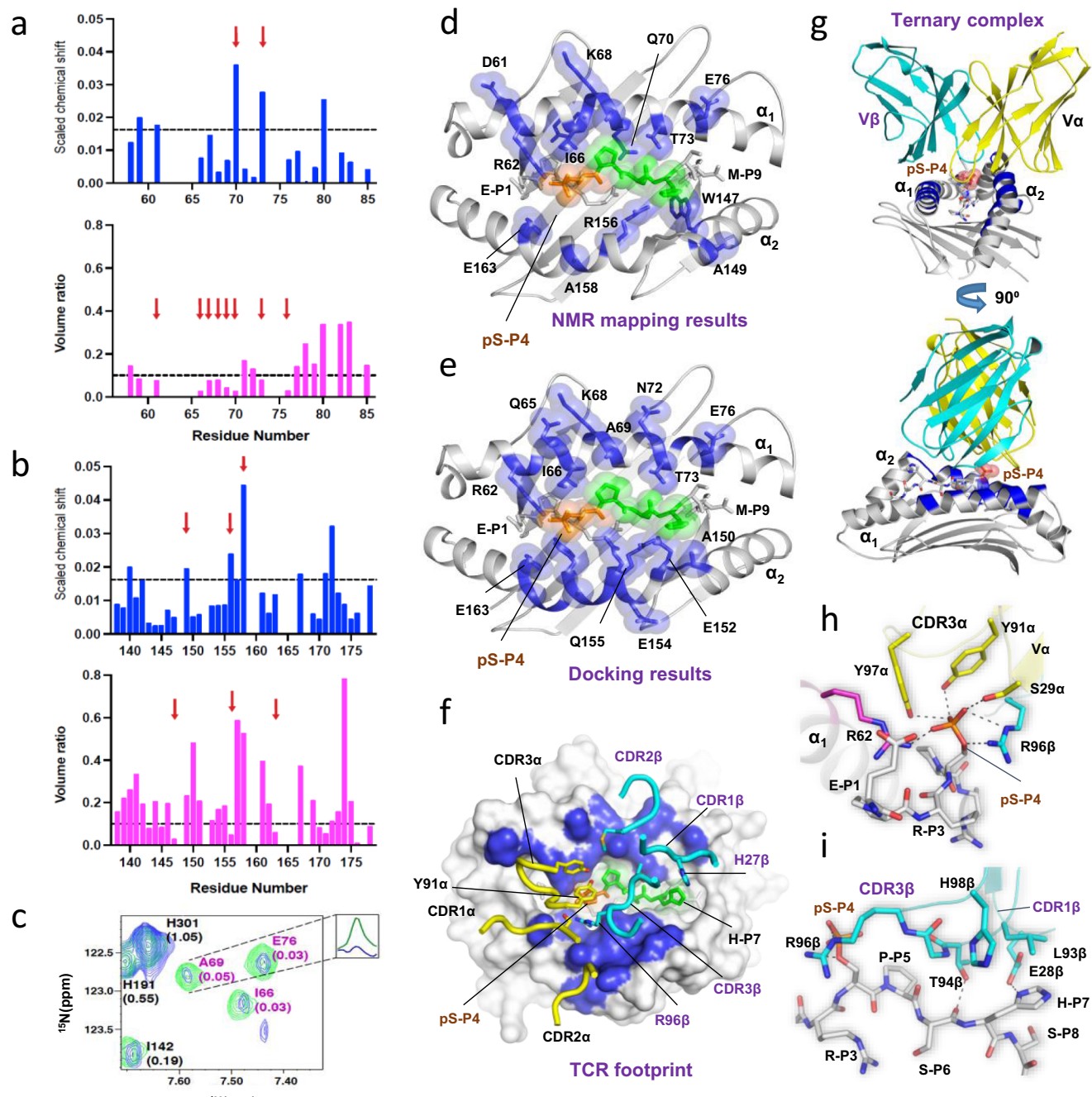

methods, such as mutagenesis and NMR[31]. We utilized 4 docking scenarios, described in Methods and schematically shown in Supplementary Fig. 8a. The pMHC interface residues were selected based on NMR mapping, peptide scan and crystallographic data, whereas the TCR residues were selected from the CDR loops based on homology with 1OGA and other pMHC-TCR structures. Selected residues and their location on the TCR_1 structure is schematically shown in Supplementary Fig. 8b. All interacting residues used in guided docking are listed in Supplementary Fig. 9a.

TCR27 adopts at least two quaternary conformations, as described above. Both structures were used to generate docking models. However, the model from the TCR_2 structure did not yield a clear solution in any of the docking scenarios based on the (1) relatively low absolute Haddock docking score for the top cluster, (2) marginal difference in score values between the top scoring cluster and the next one, and (3) a poor match with an experimentally predicted TCR-pMHC interface. By contrast, the model based on the TCR_1 structure

yielded a clear solution in each docking scenario, with the best solution set (cluster 1) having an outstanding Haddock score, the highest number of matching and common contacts within the same cluster, low interface RMSD value within the same cluster, the best geometry and the lowest clash scores (see Supplementary Fig. 10). Moreover, the top solution from each docking scenario (4 in total) with TCR_1 displayed similar topology with an RMSD value between the top solutions of 3.1 Å or less (Supplementary Fig. 8c, d). Docking was also performed using the pDOT1L$_{998–1006}$/HLA-B*0702 crystal structure as a model. The top scoring solution shared similar topology with the solution obtained for pMLL$_{747–755}$, with an RMSD value of about 1.9 Å between them (Supplementary Fig. 8e). The best docking scores were obtained in scenario #4, an example of which is shown in Supplementary Fig. 10. This assembly is discussed below.

For the assembled pMLL$_{747–755}$/HLA-B*0702/TCR27 ternary complex, there was a good match between the predicted (Fig. 7d) and observed (Fig. 7e) TCR footprint over the pMHC, which included the

**Fig. 7 | TROSY-NMR mapping of the TCR-pMHC interface and proof of a ternary complex conformation. a** NMR mapping of the HLA-B*0702 interface residues in the $\alpha_1$ helix. Top−Plot of CSP observed between $^{15}$N-TROSY spectra of pMHC (pMLL$_{747-755}$/HLA-B*0702) alone and that of a 1:1 molar ratio mixture of TCR27:pMHC. The CSP value of 0.016 indicates a threshold at 1σ over mean CSP. Bottom - Plot of ratio of peak volumes between $^{15}$N-TROSY spectra of pMLL$_{747-755}$/HLA-B*0702 alone and that of a 1:1 molar ratio mixture of TCR27:pMLL$_{747-755}$/HLA-B*0702. The line of 0.1 indicates a significance threshold value (1σ under mean volume ratio). The arrows indicate residues with significant changes in the CSP values and volume ratios. **b** NMR mapping of HLA-B*0702 residues in the $\alpha_2$ helix. Designations are the same as in (**a**). **c** Zoomed-in view of the overlay of $^{15}$N-TROSY spectra for pMLL$_{747-755}$/HLA-B*0702 (green) alone and that of a 1:1 molar ratio mixture of TCR27:pMLL$_{747-755}$/HLA-B*0702 (blue). The numbers in parentheses indicate the ratios of the peak volumes between the two spectra. Inset box shows the 1D $^1$H plane of Ala$_{69}$ in the two spectra, colored as described above. **d** HLA-B*0702 residues at the TCR-pMHC interface mapped by NMR. Cartoon model with perturbed (both CSP and peak volume change) residues (sticks and semi-transparent spheres) colored in blue. The epitope's pSer-P$_4$ and P$_5$-P$_7$ residues

(sticks and spheres) are colored in orange and green, respectively. **e** The pMHC residues (sticks and spheres) at the TCR-pMHC interface (cutoff interatomic distance <4 Å) are colored in blue. This TCR27:pMLL$_{747-755}$/HLA-B*0702 complex was generated by Haddock docking (scenario #4). **f** TCR27 footprint over pMHC in the ternary complex (scenario #4, cutoff interatomic distance <4 Å). Projections of the V$\alpha$ and V$\beta$ chains over HLA-B*0702 are shown as shaded areas, the epitope residues are sticks, and the CDR loops are coils with residues as sticks. The carbon atoms are yellow in TCR$\alpha$ and dark blue in TCR$\beta$. The carbon atoms in epitope residues P$_5$-P$_7$ are green. **g** Overall conformation of the TCR27:pMLL$_{747-755}$/HLA-B*0702 ternary complex. The second view (bottom) was obtained by rotation of the complex (top) around the vertical axis (90°). The truncated model includes only the variable region of the TCR, the peptide-binding domain of the HLA heavy chain (both are shown as cartoons) and the pMLL$_{747-755}$ peptide (stick model). The helices are labeled as $\alpha_1$ and $\alpha_2$. **h** H-bonds (dotted lines) between the phosphate group of pSer-P$_4$ and TCR27 amino acid residues (stick models) in a TCR27:pMLL$_{747-755}$/HLA-B*0702 ternary complex. **i** Interactions between TCR27 and pMLL$_{747-755}$ include atoms of amino acid residues located at the CDR3$\beta$ loop and epitope (P4–P7). Cartoon and stick representation. H-bonds are dotted lines.

epitope residues P$_4$-P$_7$ and the adjacent alpha-helical segments of HLA-B*0702. A "truncated" version of this complex including pMLL$_{747-755}$, HLA-B*0702 (domain 1) and TCR27 (variable region) is presented in Fig. 7g as a cartoon model. The TCR heterodimer binds across the longest axis of the pMHC so that the N- (P$_1$–P$_3$) and C-terminal epitope residues (P$_8$-P$_9$) remain excluded from the interface, as defined by the interatomic distance cutoff >4 Å between the non-hydrogen atoms. The shape of both (TCR27 or pMHC) interacting faces is asymmetrical, but the relative orientation of the TCR with respect to the pMHC remained similar in all 4 docking scenarios, most likely being restrained by the epitope residue selections (Supplementary Fig. 8c, d). Still, the NMR mapping and peptide scan data strongly suggest that epitope residues other than P$_4$–P$_7$ may be important but do not directly participate in recognition of TCR27. Overall, the pMLL$_{747-755}$/HLA-B*0702/TCR27 complex topology was very similar to the majority of other known TCR-pMHC-I structures[32]. The TCR27 docking angle is approximately 60° and the incident angle is 8°. The surface area buried upon complexation is approximately 2123 A$^2$. The TCR27 footprint is typical, with the V$\alpha$ domain positioned over the HLA $\alpha_2$ helix and the V$\beta$ domain positioned over the HLA $\alpha_1$ helix (Fig. 7f). The CDR3$\alpha$ loop is positioned over the epitope's N-terminus, including pSer-P$_4$, and the $\alpha_1$ HLA helix, whereas the contact area between the TCR CDR3$\beta$ loop and the pMHC includes Pro-P$_5$ and Ser-P$_6$ backbone atoms and the His-P$_7$ side chain atoms, and extends over the $\alpha_2$ HLA helix (Fig. 7f). The CDR loops maintain numerous non-covalent contacts with HLA residues such that only approximately 16% of the BSA at the TCR-pMHC interface is associated with epitope residues (Supplementary Fig. 9c, d). Despite this, the phosphate group appears to be a key interaction locus between pMHC and TCR27. Besides Arg$_{62}$ from HLA, the TCR residues contacting or close to pSer-P$_4$ include Tyr$_{91}\alpha$ and Tyr$_{97}\alpha$ (CDR3$\alpha$ loop), Arg$_{96}\beta$ (CDR3$\beta$ loop), and Ser$_{29}\alpha$ (CDR1$\alpha$ loop) (Fig. 7h). By contrast, the CDR3$\beta$ loop is positioned over, or close to, epitope residues P$_4$–P$_7$ (Fig. 7i). It is noteworthy that the key role of Arg$_{96}\beta$ (located in the CDR3$\beta$ loop) in the TCR-pMHC interaction was confirmed by site-directed mutagenesis. When the side chain of Arg$_{96}\beta$ was substituted by alanine, a mutant TCR completely lost the ability to bind the pMLL$_{747-755}$/HLA-B*0702 complex (Supplementary Fig. 9b). Importantly, all phosphate-interacting residues are hydrogen-bond donors, whereas all of the phosphate oxygen atoms are hydrogen-bond acceptors, thus providing a perfect match. The large number of ionic interactions involving all of the oxygen atoms of pSer-P$_4$ and its low solvent exposure upon complexation (less than 10% of ASA in the complex) indicates the importance of salt bridges in stabilizing the ternary complex. Overall, our data demonstrated that phosphate group integrity is vital in maintaining the pMHC-TCR complex stability, as any pSer-P$_4$ substitution (including phosphomimetics) that could alter this hydrogen-bond network affected complex stability resulting

in a loss of T cell activation and diminished binding between TCR27 and pMHC.

## Discussion

Phosphopeptide antigens are post-translationally modified (PTM) antigens that represent a new class of "non-self" (not subject to central tolerance) neoantigens associated with a broad range of cancers and virally infected cells[4,12]. Due to an underlying commonality of transformed cellular pathways, many phosphopeptide neoantigens are prevalent across patients within particular cancers and, for some phosphopeptides, across various cancers, reflecting common modes of dysregulation[6]. Thus, phosphopeptides have potential as novel targets for new immune therapies, including vaccines and cell therapies. Immunogenicity of phospho-neoantigens has been demonstrated[10,11,13,14], but their anti-cancer potential has not been explored in extensive clinical trials. In this report we describe a comprehensive SAR analysis of TCR27, discovered from healthy human donors, which recognizes an AML-linked HLA-B*0702-restricted phosphopeptide epitope pMLL$_{747-755}$. After a cross-wide genome search, we identified an additional TCR27-reactive phosphopeptide, pDOT1L$_{998-1006}$. Comparison of pMHC structures in complex with pMLL$_{747-755}$ and pDOT1L$_{998-1006}$ helped to determine the TCR footprint and the role of phosphoserine in recognition by TCR27. The cross-reactivity of TCR27 may be advantageous, as *DOT1L* is overexpressed in AML and thus is a potential therapeutic target[33]. As with pMLL$_{747-755}$, pDOT1L$_{998-1006}$ was not detected in normal tissues[34].

TCR27 avidity correlated with the affinity between TCR27 and pMHC. Our structural analysis together with binding kinetics and T cell activation data revealed that electrostatic interactions, especially between the TCR CDR3 loops and the phosphoserine-P$_4$ side chain, are crucial for the ternary pMHC-TCR complex stability. Substitution of the phosphoserine at P$_4$ with any other residue could affect geometry, size, charge distribution, and pK$_a$ values of the corresponding peptide. For instance, at neutral pH the phosphonate moiety carries a total negative charge of between −1 and −2, the sulfate carries a charge of −1, and the phosphate carries a −2 charge[35] (Supplementary Fig. 11). Loss of charge would lead to the loss of electrostatic bonds that in turn could reduce stability of the complex. From a practical standpoint, it is evident that phosphomimetics may not serve as a good substitute for phosphorylated residues in immunotherapy.

The herein described crystal structures of HLA-B*0702 in complex with distinct phosphopeptides, pMLL$_{747-755}$ and pDOT1L$_{998-1006}$, are similar to the previously solved structure of HLA-B*4002 (PDB 5IEH[19]), in terms of the relative orientation of pSer-P$_4$ of bound epitope pINCENP$_{47-55}$ (REF(pS)KEPEL), which maintained the salt bridge with the Arg$_{62}$ guanidine group despite the presence of basic Arg-P$_1$ (Supplementary Fig. 12a). By contrast, the majority of the complexes

between HLA-A*0201 and various phosphopeptides having Arg/Lys at $P_1$ displayed a distinct type of pSer-$P_4$ orientation. As an example, the structures of HLA-B*0702/ pMLL$_{747–755}$ and HLA-A*0201/pPRKD2$_{526-534}$ (RQA(pS)LSISV, PDB 3BGM[15]) are compared in Supplementary Fig. 12b–f. Notably, phosphopeptides with Arg-$P_1$ or Lys-$P_1$ were implicated in a specific orientation of pSer-$P_4$ inside the HLA-A*0201 protein binding pocket (Supplementary Fig. 8c, d and 12f). This difference is the result of a distinct AA residue composition and the charge distribution in and around the peptide-binding cavity, which is highly basic in HLA-A*0201 but more neutral in HLA-B*0702 (Supplementary Fig. 12f). The presence of Arg-$P_1$ (or Lys-$P_1$) resulted in a very high affinity between HLA-A*0201 and the vast majority of phosphopeptides tested[10,15–17]. However, in a high affinity complex the phosphate moiety was only partially exposed to solvent (Supplementary Fig. 12d, e). By contrast, in pMLL$_{747–755}$/HLA-B*0702 (and in HLA-B*04002 5IEH structure), the phosphate moiety at $P_4$ is mostly solvent exposed and thus capable of supporting an extended interaction network with the TCR. Since no other phosphopeptide-specific TCRs have been studied, it remains to be determined how the MHC-epitope binding pattern would translate in its immunogenicity and/or TCR specificity.

The incorporation of NMR data was integral to achieving successful pMHC-TCR docking. The method has previously been used to confirm complexation between various receptor/ligand pairs, though this approach differs greatly depending on the size of the complex[36], as increased molecular mass dramatically decreases the sensitivity and resolution of NMR spectra[28]. This issue can be overcome by using transverse relaxation-optimized spectroscopy (TROSY)[37], although it has not previously been applied to TCR complexes due to their sheer size and complexity. In earlier studies, truncated versions of H-2L$^d$ and 2C-TCR were used to map the TCR footprints on the MHC surface (using standard HSQC-NMR), and the results were further corroborated by the ternary complex crystal structure[38]. A similar approach was used to determine the interface between HLA and β2M[39], and to determine allosteric interactions in the CD3-TCR-pMHC complex[27,40,41]. NMR mapping is cumbersome due to spectral overlaps, but it was a key to our successful TCR-pMHC docking efforts and in validating its results. For instance, our high resolution TROSY-NMR spectral data were in good agreement with the crystallographic data and correctly pinpointed the contact residues in HLA-B*0702 interacting with pSer-$P_4$. Moreover, we demonstrated that local structural perturbations extend beyond the HLA residues that were in direct contact and include 4 main clusters of residues instead. The advantage of a cluster becomes obvious when the chemical shift for certain backbone amides cannot be assigned due to overlaps (e.g., Arg$_{62}$). Importantly, while NMR-based clustering introduced certain ambiguity in residue selection, the crystallographic data were used to more clearly define the pMHC-TCR interface by excluding from docking the HLA residues that were, for instance, solvent inaccessible or located at the HLA-β2M interface instead. As a result, TROSY-NMR in combination with crystallographic methods, mutagenesis and molecular modeling greatly improved the odds of producing a less biased and more accurate model for a ternary pMHC-TCR complex. A similar docking approach was recently applied to map the interactions between pre-TCR and MHC. The authors[42] used NMR mapping and Haddock docking to predict the tilting of pre-TCR, and the result was corroborated by the X-ray structure[43]. These studies illustrate the successful combinatorial NMR/Haddock docking approach in acquiring the complex structure.

In summary, we provided direct evidence of phosphopeptide immunogenicity by demonstrating the molecular mechanism of its recognition by a cognate TCR. These data can be utilized in further mechanistic studies and applied to the development of anti-cancer immunotherapies. Though the immunogenicity of phosphopeptides has previously been demonstrated, our work was directed to characterize a phosphoepitope-specific TCR. This enabled us to determine the molecular mechanism of interaction between the phosphopeptide, HLA and the cognate TCR, highlighting the role of phosphoserine in the epitope immunogenicity. Further, using SAR and peptide scan data we discovered the epitope pDOT1L$_{998–1006}$, which also binds to HLA-B*0702 and engages the TCR27, driving T cell effector responses similar to pMLL$_{747–755}$. Determining the process by which MHC-bound phosphopeptide antigens are presented to T cells is critical to understanding how these novel neoantigens are involved in immune recognition and tumor control, and it will aid in development of novel cancer immunotherapies.

## Methods

### Cell lines, plasmids, affinity and T cell activation assay

Tap-deficient T2 (174x CEM.T2) HLA-B*0702+ cells were received from Dr. Victor Engelhard. These and the commercially purchased Tap-deficient T2 (174 x CEM.T2) cells (ATCC CRL 1991) were maintained in RPMI 1640 (BioConcept) supplemented with 10% fetal calf serum (FCS, HyClone), 2mM L-glutamine and 1% Penicillin/Streptomycin at 37 °C and 5% $CO_2$. The EGFP reporter cell line was based on a murine TCR negative thymoma cell line derived from strain BW5417 (ATCC®TIB-47TM) and was stably transduced with (a) TCR27 in which the human constant regions were replaced with those of mouse, (b) a chimeric mouse/human CD8, and (c) an EGFP reporter construct linked to a minimal IL-2 promoter comprising three NFAT-binding sites (3xNFAT)[44]. The transduced cells were termed TCR27-AKD10R3 cells and will be called the effector cells. Mouse cell lines were cultured in SF-IMDM (BioConcept) supplemented with 3% FCS, 1% Penicillin/ Streptomycin, and 50 µM beta-mercaptoethanol at 37 °C and 10% $CO_2$.

Peptides were dissolved in dimethyl sulfoxide (DMSO) at concentrations of 2–20 mg/mL, aliquoted and stored at −80 °C. Peptide-pulsing experiments were performed by pre-incubating T2 cells with 25 µg peptide per $1 \times 10^6$ cells in PBS for 2 h at 37 °C and 5% $CO_2$, washed three times to remove unbound peptides and used for further downstream analysis. Peptide-pulsed T2 cells were incubated with effector cells in a 2:1 ratio overnight in SF-IMDM at 37 °C and 10% $CO_2$, washed twice and stained with anti-mouse TCR-® chain clone H57-597 (BD Pharmingen #560656, 1:500) for 30 min at RT. Cells were washed twice before FACS-analysis using a BD FACS Canto II with System software v2.0. Data analysis was performed using FlowJo V10 Software. Binding affinity between each peptide and HLA-B*0702 was determined using an Alpha assay. It is a homogenous, proximity-based assay for detection of peptides binding to HLA class I molecules using a conformation-dependent anti-HLA class I antibody, W6/32, as one tag and a biotinylated recombinant HLA class I molecule as the other tag, and a proximity-based signal was generated through the luminescent oxygen channeling immunoassay technology (PerkinElmer) as described[45].

### Cytotoxicity assay

Peripheral blood leukopaks were purchased from STEMEXPRESS (#CP429, de-identified donor: HLA-B*0702, BMI 20-30, female, age 20 to 50). PBMCs have been isolated using a Ficoll density gradient. Primary T cells were isolated from PBMC (EasySep Direct Human CD8+ T cell Isolation Kit, Stemcell) and transduced with a lentivirus encoding TCR27 (MOI 10:1) 24–48 h post activation with anti-CD3/CD28 beads (Dynabeads, Thermofisher) as described[22]. Cells were expanded in complete RPMI supplemented with recombinant IL-2 (100 IU/ml) (Teceleukin, Roche). Cells were challenged in the functional assays 10–15 days post initial stimulation. TCR27 expression was verified by staining cells with pMLL$_{747–755}$ -HLA-B*0702 pentamer (Proimmune) and analyzed by flow cytometry. Peptide-pulsed T2 cells were incubated with effector cells expressing or not TCR27 (1:1 ratio) in complete RPMI medium for 16 to 20 h at 37 °C and 5% $CO_2$. The procedure is shown schematically on Fig. 1A. Cells were stained for dead cells (Live/Dead, Life Technologies), and then stained with anti-CD3 (FITC

anti-human CD3 Clone UCHT1 Biolegend #300440), anti-CD4 (anti-human CD4-PerCP/Cy5.5 Clone OKT4 BioLegend #317428), anti-CD8 (anti-human CD8-PE Clone-RPA-T8, Biolegend #301051), anti-CD25 (anti-human CD25-PE/Cy7 Clone BC96, BioLegend #302612), anti-CD69 (anti-human CD69-APC Clone FN50, BioLegend #310910) monoclonal Abs (1:100 dilution) for 30 min at 4 °C, washed twice and analyzed by flow cytometry (BD Fortessa LSR II and DIVA v7.0 software). When stained for IFNγ, cells were pre-incubated for 4–6 h with BFA and Monensin (Biolegend #420601 and #420701), fixed, permeabilized and stained with PE anti-human IFNγ antibody (Biolgend #506506, 1:100 dilution). Data analysis was performed using FlowJo X software. The percentage of Target cells killing was determined as a ratio between the remaining live Target cells in each condition to the remaining live Target cells in the condition without the added T cells, amplified by 100%. The gating strategy is described in Supplementary Fig. 14.

## Identification of the off-target phosphopeptide pDOT1L$_{998-1006}$

To identify the off-target peptides we have utilized an approach that was developed previously[22]. In brief, we investigated which amino acids at each position in pMLL$_{747-755}$ enable TCR interaction by sequentially replacing every amino acid position outside of the anchor positions 2 and 9 with all 19 alternative amino acids, resulting in 134 peptides (133 altered peptides plus original epitope). Each peptide was individually evaluated using cell-based in vitro cytotoxicity assay. To represent the TCR recognition kernel, we have defined Position Weight Matrices (PWMs) for each assay by assigning normalized measurements to each of the 20 amino acids in each position. To predict potential off-target peptides, we applied an algorithm projecting the PWM-defined kernel into the human proteome, scoring TCR27 recognition for the predicted HLA-B*0702 binding 9-mer peptides. Of the 11 peptides with highest predicted cross-reactivity scores, we confirmed that only 2, pMLL$_{747-75}$ and pDOT1L$_{998-1006}$, strongly activated the TCR27-expressing effector T cells.

## Peptides

The following peptides were utilized in crystallographic and binding studies: MLL$_{747-755}$ (EPRSPSHSM), pMLL$_{747-755}$ (EPR(pS)PSHSM), E7P-MLL$_{747-755}$ (EPR(E7P)PSHSM, where E7P is (2S)-2-amino-4-phosphonobutanoic acid), OSE-MLL$_{747-755}$ (EPR(OSE)PSHSM, where OSE is (2S)-2-amino-3-sulfooxy-propanoic acid), EPRDPSHSM, EPREPSHSM, DOT1L$_{998-1006}$ (LPASPAHQL), and pDOT1L$_{998-1006}$ (LPA(pS)PAHQL. The protein sequences were derived from the following UniProt IDs: Q8TEK3 (DOT1L), and Q03164 (KMT2A).

All the peptides were synthesized using a standard Fmoc solid-phase chemical synthesis with pre-loaded polystyrene Wang (PS-Wang) resin in a Symphony-X automatic synthesizer (Gyros Protein Technologies Inc). Standard Fmoc-protected L-amino acids were activated using HCTU/NMM chemistry. For the standard phosphopeptide synthesis, the phosphate group was incorporated using N-α-Fmoc-O-benzyl-L-phosphoserine as a building block. With respect to resin substitution, a fivefold excess of amino acid, fivefold excess of activating reagent (HCTU) and a tenfold excess of N-methyl morpholine were used for coupling each amino acid. The reaction was performed for 6 min with a double coupling cycle for any incomplete coupling throughout the synthesis. These steps were repeated until the desired peptide was obtained. At the end of synthesis, the resin was washed with dichloromethane (DCM) and dried.

The sulfo-serine was produced by on-resin sulfation using dicyclohexylcarbodiimide (DCC) and $H_2SO_4$. Peptide resin, DCC, and $H_2SO_4$ were allowed to react in a fixed molar ratio of 1:5:1, respectively, in DMF at 37 °C for 10 min. The reaction vessel was drained and washed 5 times with DCM and dried. Phosphono-serine peptide was synthesized by incorporating the Fmoc-protected 2-amino- 4-phosphonobutanoic acid using the solid-phase synthesis[46]. Upon completion of peptide

assembly, the resin was transferred to another vessel for cleaving peptide from the resin. A cleavage cocktail reagent TFA: DTT: Water: TIS (88:5:5:2 v/w/v/v) was mixed with the resin and stirred for 4 h at 25 °C. The filtrate was evaporated with $N_2$ gas followed by precipitation with chilled diethyl ether and stored at −20 °C for 12 h. The precipitated peptides were centrifuged and washed twice with ether, dried, dissolved in the choice of solvent, and lyophilized to produce a crude dry powder.

The peptides were purified by HPLC on a Luna C$_{18}$ column (Phenomenex, Inc) using a water (0.1% TFA) vs acetonitrile (0.1% TFA) gradient. Peptide purity and stability was tested using analytical Luna C$_{18}$-column (Phenomenex, Inc). Peptide sequences were confirmed by LC/MS (6550 Q-TOF, Agilent Technologies).

## Protein sequences

Amino acid sequences for human HLA-B*0702 (fragment, amino acid residues 25–299, Genebank ID AZU90178) and Human β2-microglobulin (hβ2M), a fragment with residues 21–119, UniProt ID P61769) were selected for expression in Escherichia coli. Another version of recombinant HLA-B*0702 also included an AVI-tag sequence (GLNDIFEAQKIEWHE)[47] added at the protein C-terminus. TCRα and TCRβ chains included variable domains for TRAV27 (Genebank ID MZ701715, AA residue 1–110) and TRBV27 (Genebank ID MZ701716, AA residues 1-110), respectively, that were linked to the corresponding human LC13 constant domains[48]. To further stabilize recombinant TCRαβ heterodimer and improve protein yield during expression in E. coli, an inter-chain disulfide bridge was introduced between the constant chains, as described previously[48].

## Gene cloning, protein expression, refolding and purification

Ectodomains for human HLA-B*0702, hβ2M, or TCRα and TCRβ chains were cloned and expressed individually as inclusion bodies according to the previously described protocols[27,49] with minor modifications. In brief, a codon-optimized cDNA for every gene was synthesized by Genewiz, cloned between the NdeI and HindIII restriction sites of a plasmid pET30a, and verified by DNA sequencing. Mutagenesis was performed using Quick Change II Site-Directed Mutagenesis kit (Agilent) according to the manufacturer's instruction, and the results were verified by DNA sequencing. After transformation into E. coli BL21(DE3), the exponentially grown bacteria were placed in LB+Kanamycin (50 µg/ml) medium, protein expression was induced by adding IPTG (Sigma) to the final concentration of 1 mM, and the cells were incubated by shaking for 4–5 h at +37 °C. The cell pellet was harvested by centrifugation, and the proteins as inclusion bodies were purified as described earlier[27], then solubilized in a buffer containing 6 M Guanidine-HCl, 50 mM Tris-HCl (pH 8.0), cleared by centrifugation (15,000 g, 10 min), aliquoted and kept frozen at −80 °C until use.

Refolding of denatured proteins was performed as described[27,48,49] and isolated by gel-filtration. Each pMHC complex with a different peptide was refolded separately. Purified proteins were dialyzed against 10 mM HEPES-NaOH (pH 7.5), protein purity was validated by non-reducing SDS-PAGE (4–20%) and mass-spectrometry. Proteins were concentrated to 5-10 mg/ml, and either flash-frozen in liquid nitrogen and stored at −80 °C until use or subjected directly to crystallization and other analyses. TCRα and TCRβ were cloned and expressed separately. The TCR heterodimer was refolded in conditions, as described above for pMHC, but using a 1:1 molar ratio between the TCR subunits. Additionally, the refolding buffer was supplemented with 3 M urea. For long-term storage, proteins were frozen in liquid nitrogen and kept at −80 °C.

## Biolayer interferometry

The AVI-tagged version of recombinant HLA-B*0702 was used to refold the pMHC in complex with each peptide separately. Refolded complexes were purified by gel-filtration on Superdex 200 (Cytiva).

Recombinant BirA enzyme was expressed in *E. coli* and purified in house, as described earlier[50]. The BirA500 biotinylation kit (Avidity) was used to biotinylate the proteins according to the manufacturer's instruction, but using in-house produced BirA. Biotinylated pMHC were purified by gel-filtration on Superdex 75 (Cytiva) using HST buffer without BSA. Protein biotinylation was confirmed by the gel-shift assay with streptavidin (Fisher Scientific) after biotinylation and immediately prior to the BLI experiment. An example of the gel-shift assay is shown in Supplementary Fig. 13. Only the completely biotinylated proteins were used in the study.

Binding between pMHC and TCR27 was measured using a BLI Octet RED96 instrument and streptavidin-conjugated SA chips according to the manufacturer's (Pall ForteBio, Menlo Park, CA) instructions and as previously described[51,52]. In brief, HST buffer (10 mM HEPES, pH 7.5, 100 mM NaCl, 0.01% Tween-20, 1% BSA) was used in all analyses. Experiments with different pMHC complexes were performed in parallel on the same day and using the same set of conditions to eliminate a batch effect. Each biotinylated pMHC was diluted to 2–10 µg/ml, immobilized on a sensor chip with approximately the same signal magnitude for each peptide (-1 nm), and equilibrated in HST buffer for 1 min. It was then followed by the TCR association step (30 s) in 200 µL of TCR27 solution (TCR27 concentrations were ranging between 0 µM and up to 20 µM or 50 µM), and the dissociation step in 200 µL of HST buffer for 3–20 min. Assays were performed at 25 °C. Data were analyzed using Octet® 9.1 System Data Analysis software to determine dissociation constant values.

## Protein crystallization, structure solution and refinement

The proteins were crystallized using sitting drop vapor diffusion methods and the 1:1 (vol:vol) protein to precipitant ratio. To identify initial conditions, the MCSG crystallization screens have been utilized (Anatrace, OH, USA). The crystallization drops were set up using the 96-well Intelli-plates (Art Robbins Instruments) and a Mosquito crystallization Robot (SPT Labtech), and the plates were kept at +18 °C. The crystals appeared between 3–10 days and up to 6 months from the beginning of the incubation. Crystallization conditions were optimized manually. After optimization, each pMHC complex was crystallized by mixing 0.5 µl of protein solution (8–10 mg/ml) with 0.5 µl of precipitant A (18–26% PEG 4000, 0.1 M sodium citrate, pH 8.0, and 20% isopropanol) in the 24-well Intelli-plates (Art Robbins Instruments) and equilibrated against the same precipitant at +18 °C. The rod-like crystals appeared 2–10 days after beginning of incubation and grew up in length of up to 1.5 mm. The crystals were flash-frozen in liquid nitrogen ($LN_2$) after brief soak in the precipitant solution supplemented with 20-30% (vol/vol) of a cryoprotectant such as glycerol, ethylene glycol or sucrose, and stored frozen in $LN_2$ until data collection. In addition, the $pMLL_{747-755}$/HLA-B*0702 pMHC complex was crystallized using a precipitant B, such as 30–35% PEG 4000, 0.1 M Tris-HCl, pH 8.5, 0.2 M sodium acetate. In these conditions the prism-shaped crystals appeared in 3–6 months and grew up to the average size of around 100 microns; they were directly flash-frozen and stored in $LN_2$ until collection of the X-ray diffraction data.

The diffraction quality crystals for TCR27 were obtained by the crystallization method described above and using the following set of precipitants, (1) 20–25% PEG 1000, 50 mM HEPES-NaOH (pH 7.5), crystal morphology−plates; and (2) 20–25% PEG 4000, 0.2 M ammonium sulfate, 0.1 M sodium citrate (pH 5.6), crystal morphology−prisms and plates. In conditions #1 the crystals grew up in 5–10 days, but in conditions #2 they appeared after 3–4 months. The crystals were frozen directly in $LN_2$ and stored frozen until further analysis.

The X-ray diffraction data were collected from the frozen crystals (100°K) at the APS beamline 19BM (using SBCCollect) or at the NSLS-II beamline 17-ID-2 (FMX) (using LSDC software). The data were integrated and processed by HKL3000 (19BM) or XDS (FMX), respectively,

and scaled by AIMLESS/CCP4 package[53–55]. The crystal structure for HLA-B*0702 in complex with $pMLL_{747-755}$ peptide was solved by molecular replacement (MR) using PHASER[56] and the PDB structure 5WMN as a search model, and then refined by REFMAC[57] and finally, fixed manually using COOT 0.9.7 visualization and refinement software[58]. The refined structure for $pMLL_{747-755}$ /HLA-B*0702 complex served as a search model to solve other pMHC structures described in this report. The TCR27 crystal structure was solved by MR using the PDB file 1OGA as a search model and refined as described above. All the models and the corresponding structure factor files have been validated using Sfcheck and Procheck software in CCP4, validation tools embedded in COOT (including Molprobity), and the validation server at the PDB. The X-ray data collection and refinement statistics are presented in Supplementary Table 1 altogether with PDB IDs for all the deposited structures. When the precipitant A was utilized in crystallization of pMHC complexes with different peptides, it resulted in isomorphous crystals.

The structure-based pairwise alignment between two sets of atomic coordinates was carried out using a PYMOL align algorithm (or super for non-identical structures) and zero refinement cycles or "LSQ Superpose" command in COOT. Sequence alignment was performed by NCBI BLAST (https://blast.ncbi.nlm.nih.gov/Blast.cgi). Output data from PYMOL alignment were utilized to produce the CA-CA difference plots in MS Excel. To identify the actual (and potentially relevant to epitope binding) differences between the two closely related structures, we had to minimize the effect of "random" structural heterogeneity, related to crystal packing, thermal residue motion and experimental errors. To do this, we utilized a "core RMSD" value (in addition to RMSD), which was calculated by using the command align in PYMOL, coordinates of all atoms and up to 5 refinement cycles upon condition that the number of rejected outliers does not exceed 20% of the total atom count. The reason behind this number was the fact that around 40 AA residues of the HLA domain 1 (residues 1–180) are located in the outer loops, which comprise -20% of all the residues for this domain. For isomorphous crystal structures, the core RMSD value remained very similar to the experimentally determined coordinate errors, varying between 0.1 Å and 0.2 Å for all the structures. Solvent-exposed surface area (SASA, $Å^2$) for individual amino acid residues was calculated using an EBI PISA software[59] or in PYMOL using a get_sasa_relative command, and was expressed as a percentage out of all residue surface area (ASA). The sigmaA-weighted 2Fo-Fc or Fo-Fc electron density maps in CCP4 format for drawing were generated in COOT[58] using amplitude (FWT) and phase (PHWT) data from the REFMAC mtz output structure factor file. Structural figures were produced using PYMOL 2.4, Microsoft Excel (MS Office 2019) and Biorender[60].

## Triple-resonance experiments for NMR backbone assignments

NMR analysis of large protein molecules is always confounded by its size, asymmetry and dynamical nature[28]. To overcome such complexities and resulting signal relaxation that leads to poor spectra, HLA-B*0702 molecules were prepared in a 100% deuterated minimal media with either $^{15}N$ or $^{15}N$ plus $^{13}C$ labels and refolded with unlabeled hβ2M and peptide as follows.

Triple labeled HLA-B*0702 ($^{15}N$, $^{13}C$, $^2H$) was produced as inclusion bodies in *E. coli* by supplementing 900 ml of M9 minimal media with 1 g of $^{15}NH_4Cl$, 2 g of $^{13}C,^2H$-glucose, 1 g of $^2H,^{15}N,^{13}C$-celtone (Cambridge Isotopes) and 100% $D_2O$[61]. Protein production was induced with 1 mM IPTG at culture $OD_{660} = 0.6$ and then continued for 4 h at 37 °C after that. Bacteria were pelleted by centrifugation, the inclusion bodies were purified and used in pMHC refolding along with unlabeled hβ2M and peptide, respectively. Refolded pMHC was purified by gel-filtration in PBS, pH 7.4, as mentioned above. The buffer in a protein sample was exchanged with 25 mM MES, 150 mM NaCl, pH 6.5. A final pMHC concentration of 200 µM was used in NMR experiments.

TROSY-based triple-resonance experiments were performed as suggested[37] on the perdeuterated,$^{15}$N,$^{13}$C pMLL$_{747-755}$/HLA-B*0702 sample at 30 °C on 600 MHz (Bruker AVANCE III, New York University) and 800 MHz (Bruker AVANCE III HD, New York Structural Biology Center) spectrometers, equipped with cryoprobes to obtain the backbone atoms N, HN, Cα, C′ assignments. The NMR data were processed by NMRPipe 10.9 Rev 2021.139.10.46[62] and analyzed using NMRViewJ 9.2.0-b27[63]. Each peak in the $^{15}$N-TROSY spectra corresponded to one amide (N–H) pair belonging to a protein amino acid. About 86% of all assignable (excluding proline residues) backbone N and HN atoms were assigned. Further, about 84% of the alpha-helical residues that interact with pMLL$_{747-755}$ peptide and HLA were assigned (Supplementary Data 1). The assignments were transferred without ambiguity to the $^{15}$N-TROSY spectra of pMLL$_{747-755}$/$^{15}$N,$^{2}$H-HLA-B*0702 obtained in HST buffer containing 10 mM HEPES, 100 mM NaCl, 0.1% P-20 (surfactant), pH 7.5. The same buffer was used in the BLI experiments. NMR assignment data were deposited to the Biological Magnetic Resonance Data Bank, BMRB (deposition ID 51815).

### Production of pMLL$_{747-755}$/HLA-B*0702 with labeled amino acids

The pMLL$_{747-755}$/HLA-B*0702 complexes with $^{15}$N-Arg, $^{15}$N-Leu or $^{15}$N-Tyr labels were prepared by growing *E. coli* expressing HLA-B*0702 in 500 ml of M9 minimal media containing unlabeled ingredients including 500 mg unlabeled amino acids except for 50 mg individual $^{15}$N-labeled amino acids listed above. Each complex was refolded with unlabeled hβ2M and unlabeled pMLL$_{747-755}$ peptide and purified as mentioned above. These protein samples were used to resolve NMR assignment ambiguities and peak overlaps.

### TROSY-NMR mapping of HLA-epitope interface

Perdeuterated ($^{2}$H), $^{15}$N-labeled HLA-B*0702 was produced as inclusion bodies in *E. coli* by supplementing 900 ml of M9 minimal media with 1 g of $^{15}$NH$_4$Cl, 2 g of $^{2}$H-glucose, 1 g of $^{2}$H, $^{15}$N-celtone (Cambridge Isotopes) and 100% D$_2$O.Perdeuterated ($^{2}$H), Protein production was induced with 1 mM IPTG when culture OD$_{660}$ was 0.6 and incubation continued for 4 h at 37 °C after induction. Each complex was produced by refolding labeled HLA-B*0702 in the presence of unlabeled hβ2M and unlabeled pMLL$_{747-755}$ or unlabeled MLL$_{747-755}$ peptide, respectively, and purified as mentioned above. The buffer in protein samples was exchanged into HST buffer, and the final protein concentrations were of 216.4 μM (pMLL$_{747}$/755-HLA-B*0702) and 200 μM (MLL$_{747-755}$/HLA-B*0702), respectively. $^{15}$N-TROSY data were collected using a 600 MHz NMR spectrometer at 25 °C. Assignments were transferred from the $^{15}$N-TROSY spectra of pMLL$_{747-755}$/HLA-B*0702 to $^{15}$N-TROSY spectra of MLL$_{747-755}$/HLA-B*0702. The scaled (normalized) chemical shift differences between the HLA residues in both spectra were calculated using the formula, $\delta_{ppm} = ((\delta H)^2 + 0.11(\delta N)^2)^{1/2}$ as in[27]. Residues with significant CSP values were defined to have the chemical shift difference ($\delta_{ppm}$) greater than 1σ above the mean value.

### TROSY-NMR mapping the pMHC-TCR interface

Perdeuterated α and β subunits of TCR27 were produced separately as inclusion bodies in *E. coli* by using M9 minimal media containing $^{2}$H-glucose, $^{2}$H-celtone (Cambridge isotopes) and 100% D$_2$O. Protein expression was induced with 1 mM IPTG when culture OD$_{660}$ was 0.6, and continued for 4–5 h at 37 °C after that. Cells were collected and inclusions bodies isolated, the TCRαβ heterodimer was refolded and purified as mentioned above. The deuterated TCR27 sample was exchanged into HST buffer for chemical shift perturbation experiments.

Titrations of perdeuterated TCR27 with perdeuterated pMLL$_{747-755}$/$^{15}$N-HLA-B*0702 was performed on a 600 MHz

spectrometer (Bruker AVANCE III) at 25 °C. $^{15}$N-TROSY was collected for samples with the following pMLL$_{747-755}$/HLA-B*0702 and TCR27 ratios, 100 μM:0 μM (1:0), 93 μM:23.4 μM (1:0.25), 88 μM:44 μM (1:0.5) and 78.5 μM:78.5 μM (1:1), respectively. Another $^{15}$N-TROSY spectrum was collected for the binary pMLL$_{747-755}$/HLA-B*0702 complex alone at 78.5 μM concentration for CSP and peaks intensity analyses. Residues with overlapping spectral peaks were excluded from calculations and subsequent analyses. Assignments were carefully transferred from the pMHC (pMLL$_{747-755}$/HLA-B*0702, 78.5 μM) spectrum to the TCR27 plus pMHC (78.5 μM:78.5 μM) spectrum. The scaled chemical shift differences ($\delta_{ppm}$) were calculated using the same formula as in the previous section. Residues with significant CSPs were defined to have the chemical shift difference ($\delta_{ppm}$) greater that 1σ above the mean value. Residues with significant signal intensity change were with a peak relative volume (V/V$_0$) value higher than 1σ below the mean value. The NMR spectral data and the differential analyses are in the Supplementary data 2.

### TCR-pMHC docking using Haddock 2.4

The protein–protein information-driven docking between pMHC and TCR27 was performed using HADDOCK software[31]. Docking models were generated using the HADDOCK webserver (https://wenmr.science.uu.nl)[64] version 2.4, using docking scenarios which differed by selection of interacting amino acid residues at the TCR-pMHC interface (see below). In each docking experiment the following settings were used: top scored 1000 models were selected at the rigid-body (it0) stage and the top scored 200 models for the flexible (it1) and water refinement steps. The random removal of restraints was set to 50% for each docking run. The interacting residues were specified as either active or passive residues (such as directly involved in TCR-MHC interface or indirectly – as neighboring residues, respectively) to estimate the docking bias. The pMHC residues were selected using the data obtained by site-directed mutagenesis, peptide scan, NMR and crystallographic data described in this report. The TCR27 residues were selected on the CDR loops only. In every Haddock run, TCR27 was receptor, and pMHC was a ligand. The initial models for docking were generated from the crystal structures produced in this study, including the two TCR27 conformations, TCR_1 and TCR_2, respectively (see Supplementary Table 1). To reduce running time, every pMHC model was truncated to include the HLA-B*0702 domain 1 (AA residues 1-180) in complex with corresponding epitope (pMLL$_{747-755}$ or pDOT1L$_{998-1006}$, respectively). Each truncated TCR model included the variable Vα and Vβ domains only (AA residues 1-110 from each TCR chain). The following interface residues were selected in the docking scenarios 1–4: #1—the epitope residues P$_4$–P$_7$ and the CDR3 loops, #2—the epitope residues P$_4$–P$_7$ and the CDR loops, #3—the epitope residues P$_4$–P$_7$, NMR-mapped HLA residues (with SASA > 20%) and the CDR3 loop, #4—the epitope residues P$_4$–P$_7$, NMR-mapped HLA residues and all CDR loops. Further details are described in the results section. Top four clusters with highest HADDOCK scores from each run were analyzed, and the top scoring clusters from every scenario were compared between each other to validate the docking strategy and identify the most correct TCR-pMHC model.

### Reporting summary

Further information on research design is available in the Nature Portfolio Reporting Summary linked to this article.

## Data availability

The data that support the findings of this study are openly available in the PDB at https://www.rcsb.org, new structures reported in this study: 7S8J, 7S8I, 7S7E, 7S7F, 7RZD, 7RZJ, 7S79, 7S7D, 7S8A, 7S8E, 7S8F. Previously reported structures used here: 1OGA, 5IEH, 3BGM, 5WMN, NMR assignment data are available in the Biological Magnetic Resonance Data Bank, BMRB: BMRB 51815. The sequences for TCR27 have been

deposited to Genebank under IDs MZ701715 and MZ701716. Other data supporting the findings of this study are available within the article and its supplementary materials. Source data are provided with this paper.

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

## Acknowledgements

We thank Duane Moogk (McMaster University) for critical reading of the manuscript. This work was supported by the NIH grant NIGMS R01 GM124489 (to M.K.), NCI R01 CA243486 (to M.K), New York University Melanoma SPORE P50CA225450 (to M.K.), the New York University Center for Blood Cancer (CBC) Pilot Fund grant (to M.K. and Y.P.), and a Sponsored Research Agreement from Agenus to M.K. The NMR spectrometers at the NYU Chemistry Shared Instrumentation Facility were supported by NYU and the NIH Grant 1S10-OD016343. The facilities at the NYSBC were supported by the NIH Grant P41GM118302. Mass spectrometry experiments for protein identification were supported in part by NYU Langone Health and Laura and Isaac Perlmutter Cancer Center support grant P30CA016087 from the National Cancer Institute. Results shown in this report are partially derived from work performed at Argonne National Laboratory, Structural Biology Center at the Advanced Photon Source. SBC is operated by UChicago Argonne, LLC, for the U.S. Department of Energy, Office of Biological and Environmental Research under contract DE-AC02-06CH11357. Results in this report are partially derived from work performed at The Center for BioMolecular Structure (CBMS) primarily supported by the National Institutes of Health, National Institute of General Medical Sciences (NIGMS) through a Center Core P30 Grant (P30GM133893), and by the DOE Office of Biological and Environmental Research (KP1607011). As part of NSLS-II, a national user facility at Brookhaven National Laboratory, work performed at the CBMS is supported in part by the U.S. Department of Energy, Office of Science, Office of Basic Energy Sciences Program under contract number and DE-SC0012704.

## Author contributions

Experiments were conceptualized by Y.P., A.N., L.P., X.M., D.U. and M.K. D.U. and M.K. supervised the whole project. X-ray crystallography sample preparation, data collection, structure refinement and analysis were per-formed by Y.P., L.P., and S.N. NMR sample preparation, data collection and analysis were performed by A.N. and Y.P. Biolayer interferometry sample preparation, data collection and analysis were performed by Y.P., L.P. and S.N. Site-directed mutagenesis was performed by S.N. Haddock docking and analysis were performed by Y.P. and L.P. The T-cell related studies and peptide synthesis: B.J., B.M., C.B., O.H., S.G., F.S. (investigation); T-SPRINT data analysis: A.S.Y. (investigation); B.J., B.M., C.B., O.H., S.G. (data analysis and validation); B.J., B.M., C.B., F.S., M.A.F., J.S.B., A.H., M.v.D. (resources); B.M., C.B. (visualization); B.J., B.M. and M.A.F. (supervision); R.B.S. provided critical insight and advice during the studies; X.M., D.P. and E.C. con-ceived, designed and performed the cell-based experiments. The draft was written and edited by Y.P., A.N., L.P., B.J., B.M., D.U. and M.K.

## Competing interests

All authors have directly participated in the planning, execution, or analysis of the study and drafting or revising of the manuscript, and have read and approved the final version of the manuscript. Y.P., L.P., S.N., A.N. and M.K. have no financial interests to disclose. D.U., B.M., X.M., B.J., C.B., O.H., S.G., E.C, A.S.Y., F.S., R.B.S., M.A.F., A.H., D.P., M.v.D., J.S.B. have ownership of equity securities and/or are currently or previously employed by Agenus. This does not alter our adherence to policy of the journal on sharing the data and materials.
