## [Peer review file · Nature Communications]

REVIEWER COMMENTS

Reviewer #1 (Remarks to the Author):

In the paper, authors present a rather extensive study of an HLA-B*0702 restricted AML phosphoantigen pMLL747-755 and its cognate TCR, TCR27 by using x-ray crystallography, NMR, mutagenesis, and modeling etc. Their key discovery is that the replacement of phosphoserine P4 with either serine or phosphomimetics weakens the binding between pMHC and TCR and abolishes T cell activation as well. They also identified a cross-reactive phosphopeptide pDOT1L998-1006 and performed some parallel experiments. The amount of work covered in the paper was extensive. However, there are numbers of issues that prevent a recommendation to publish the work in Nature Communications. The major problem is that in this structure heavily weighted paper, authors are seemingly not familiar with some fundamental aspects of structural biology, particularly in the area of pMHC-TCR complex. The following are only some of the issues or problems.

1. Except the complex structures of HLA-B*4001 and HLA-DR1 with phosphopeptides, the structures of three HLA A2-restricted phosphopeptides published in 2009 should have been mentioned in the introduction. This paper was published by J.Petersen etc. with a PMID of 19196958. Additionally, it is worthwhile to compare some of them to those studied in this work.

2. Problems in scientific terms.

For example, in Page 5, the sentence “serine hydroxyl group was replaced by phosphonate: (2S)-2-amino-4-phosphonobutanoic acid)” is confusing. It should be said either “serine hydroxyl group was replaced by phosphonate group” or “serine is replaced by (2S)-2-amino-4-phosphonobutanoic acid”. The sentence following this one in text has the same problem.

Authors use polar bond to describe hydrogen bond or salt bridge in many parts of the manuscript. Polar bond is usually defined as a type of covalent bond. Neither a hydrogen bond or a salt bridge is a polar bond.

RMSD and RMSD plot. RMSD, root-mean-square deviation, is a value resulted from a structural alignment, representing the average distance between paired atoms. In the text, authors should clarify how each alignment such as overall alignment was performed but also explain how core RMSD was measured. Most importantly, those so-called RMSD plots, such as Figure 5D-G and some in supplementary figures, are not RMSD plots. They are in fact atom-atom distance plot after an alignment.

“2FoFc electron density map” should be “2Fo-Fc electron density map”. This may be trivial. It should be also mentioned if the map is weighted. If so, how it was weighted.

In the pMHC-TCR community, when a pMHC structure is presented, a common practice is to orient pMHC in such a way that the N-terminus of peptide is on the left and the C-terminus is on the right. Of course, authors certainly have their own ways to orient pMHC in a figure. But, in one figure, its orientation should be consistent. It is very confusing to orient one in one way and reverse the orientation in another, even for two parallel figures. Check Figure 4,6,7 and related supplementary figures.

3. Problems in scientific conception. There are number of places in the text that are full of misconception and confusion. For example, on Page 21, the paragraph on TCR is very problematic.

“In every TCR, the complementarity determining regions (CDRs) responsible for epitope-MHC recognition are located at the extremities (CDR loops) of the variable (V α and V β) domains (Rudolph et al., 2006).”

What is “epitope-MHC” recognition?

It sounds like the “CDRs” are CDR loops. The word “extremities” used here to describe CDR loops is incorrect and unprofessional.

Do authors know the exact locations of CDRs in the V domain?

“The CDRs in TCR27 were defined based on similarity with other TCRs, ...”

Which other TCRs were used for the definition?

It is not a correct way for CDRs definition.

“The TCR27 structure is an $\alpha\beta$ heterodimer, where each monomer consists of one constant and one variable domain, all arranged into the constant and the variable regions, respectively”

Very confusing.

“Unexpectedly, the quaternary conformation of TCR_1 was different from that of TCR_2, with an RMSD value between the coordinates of their C α atoms of 3.1Å.”

There is something totally wrong here. It is impossible to have such a huge RMSD value from an alignment of two conformations of one TCR. The authors need to understand the meaning of RMSD first. It is a term used in many places of the paper, such as in RMSD plots as commented above.

“The “hinge” areas around which the subunit “rotation” occurs (shown by the arrows in Fig.5D and Fig.5G) is located at or close to the dimer interface between the two variable TCR domains.”

What is the meaning of “hinge areas”, which is located at or close the dimer interface?

4. Some potential technique issues.

Modeling issue. It is fine to make a prediction of a TCR-pMHC interaction model without their complex structure. But some general knowledges of the interaction between TCR and pMHC are required to judge the quality of a model. If in a model TCR interacts only to the peptide without important participation of MHC, the model is wrong.

Water molecules. All eleven crystal structures reported in this paper have resolution limits between 2.10 Å and 1.53 Å. Water molecules could be easily identified in these structures. It is well known that water molecule plays an important role in the formation of a hydrogen bond network between peptide and MHC. Strangely, authors never mentioned any water molecules in the analysis of these structures.

R factors in refinement. All eleven crystal structures in different resolution ranges have nearly the same R and Rfree factors, R=16.74(+/-)0.38% and Rfree=21.09(+/-)1.12%. Maybe, it is what truly happened. It is very rare.

Reviewer #2 (Remarks to the Author):

The paper “Molecular mechanism of phosphopeptide neoantigen immunogenicity” is an excellent study of the characterization of an $\alpha\beta$ TCR interaction with phosphopeptide MHC complex in comparison to related pMHC. The authors thoroughly explore peptide binding space surrounding the target epitope using biophysical and functional techniques to obtain a convincing understanding of the binding events in question. There is good cross-fertilization and concordance between the various technologies that is supportive of the models presented. Overall, this is a well-designed study that is worthy of publication. There are some minor concerns that should be easily addressed.

1. The NMR assignments should be deposited in the Biological Magnetic Resonance database (BMRB) <https://bmr.io/>. If already deposited, then the accession number should be listed.
2. The range of variation between HADDOCK docking overlays in Fig. S7 would be clearer if you could show a top down view of the highest scoring models like you show in Fig. 7G showing the differences in

positioning of the six CDR loops or at least the CDR3s if that makes the figure clearer. Alternately it would be helpful to include a supplemental PDB file (not for deposition, but just for reader evaluation). As a reader, I would be very interested to have access to this model. The figure showing the residues used in the HADDOCK docking is very helpful but the residues should also be enumerated as simple text in the legend or a separate table.

3. It would be remiss not to mention the parallels your study has to a recent series of publications on the preTCR-pMHC interaction wherein X-ray structures were used in interpreting data from NMR to model a complex using HADDOCK(Mallis, et al J Biol Chem 2018, PMID29101227). The conclusions of that study were initially less direct due to the weak nature of the interaction, nevertheless, the followup studies did lead to actionable data and also to an X-ray structure (Mizsei, et al J Biol Chem 2021 PMID33837736 ; Li, et al Science 2021 PMID33335016). It is at your discretion whether to add mention of this in the discussion.

Reviewer #3 (Remarks to the Author):

The present manuscript “Molecular mechanism of phosphopeptide neoantigen immunogenicity” is a detailed and well laid-out study that nicely brings together biophysical, structural and cellular tools and approaches to gain additional mechanistic understanding of phosphopeptide neoantigen immunogenicity. The choice of orthogonal methods (e.g. NMR, Structure, Docking, Cellular assays) has been done very careful each adding valuable information to get a holistic mechanistic view on how phosphopeptide antigens are contributing to a immunogenic response.

A couple of minor suggestions to further improve the quality and clarity of the manuscript:

(1) In the Methods part you describe the use of biotinylated pMHC for the BLI studies. Please add how this protein material was prepared (in vivo vs. in vitro biotinylation ?), including the level of biotinylation as assessed by MS.

(2) In the Methods part you describe the E.coli medium used for the preparation of the triple labeled HLA-B*0702 protein. Please state the concentrations used for $^{15}\text{NH}_4\text{Cl}$ and $^{13}\text{C}_2\text{H-glucose}$ as well as the $^2\text{H},^{15}\text{N},^{13}\text{C}$ -celtone.

(3) BLI was employed yo characterize the physical interactions between pMHC ligands and TCR27, which clearly showed that the binding affinity between pMHC and TCR27 is dependent on the nature of the amino acid residue at P4. For transparency it will be important to share the raw data of the sensorgrams in the main manuscript together with the associated equilibrium-binding data fits (plus kinetic fits for

interactions that have clear kinetics features that can be analyzed). Particular the kinetics values can further our understanding of the molecular interactions that are taking place. Furthermore it will be important to add error bars (SEM) on Figure 2A as they are not visible in the plot but SEM values are given when presenting the affinity values.

I would like to congratulate the authors for the excellent work, the nice study outline and clarity of the results and conclusions.

We would like to thank the reviewers for their insightful and constructive comments and valuable suggestions. We believe that we have addressed all of the specific issues that were raised, and the relevant changes have been incorporated into the revised manuscript and highlighted in yellow. The answers to reviewer's comments are provided below and highlighted in yellow.

Point-by-point response to the reviewers' comments, reproduced verbatim

Reviewer #1 (Remarks to the Author):

In the paper, authors present a rather extensive study of an HLA-B*0702 restricted AML phosphoantigen pMLL747-755 and its cognate TCR, TCR27 by using x-ray crystallography, NMR, mutagenesis, and modeling etc. Their key discovery is that the replacement of phosphoserine P4 with either serine or phosphomimetics weakens the binding between pMHC and TCR and abolishes T cell activation as well. They also identified a cross-reactive phosphopeptide pDOT1L998-1006 and performed some parallel experiments. The amount of work covered in the paper was extensive. However, there are numbers of issues that prevent a recommendation to publish the work in Nature Communications. The major problem is that in this structure heavily weighted paper, authors are seemingly not familiar with some fundamental aspects of structural biology, particularly in the area of pMHC-TCR complex. The following are only some of the issues or problems.

1. Except the complex structures of HLA-B*4001 and HLA-DR1 with phosphopeptides, the structures of three HLA A2-restricted phosphopeptides published in 2009 should have been mentioned in the introduction. This paper was published by J.Petersen etc. with a PMID of 19196958. Additionally, it is worthwhile to compare some of them to those studied in this work.

We thank the reviewer for the suggestion. The paper by J. Petersen et al. (2009) was included. This and other relevant publications referenced in the manuscript helped to explain how the nature of the amino acid residue at P₁ affects the conformation of phosphoserine P₄ when the phosphopeptide is bound to HLA-A*0201. The corresponding paragraphs were added to the Introduction and Discussion sections.

2. Problems in scientific terms.

For example, in Page 5, the sentence "serine hydroxyl group was replaced by phosphonate: (2S)-2-amino-4-phosphonobutanoic acid)" is confusing. It should be said either "serine hydroxyl group was replaced by phosphonate group" or "serine is replaced by (2S)-2-amino-4-phosphonobutanoic acid". The sentence following this one in text has the same problem.

We agree with the reviewer, and we revised the sentences accordingly.

Authors use polar bond to describe hydrogen bond or salt bridge in many parts of the manuscript. Polar bond is usually defined as a type of covalent bond. Neither a hydrogen bond or a salt bridge is a polar bond.

We agree with the reviewer, and we removed the term "polar bond" from the text and replaced it with the correct term (H-bond).

RMSD and RMSD plot. RMSD, root-mean-square deviation, is a value resulted from a structural alignment, representing the average distance between paired atoms. In the text, authors should clarify how each alignment such as overall alignment was performed but also explain how core RMSD was measured. Most importantly, those so-called RMSD plots, such as Figure 5D-G and some in supplementary figures, are not RMSD plots. They are in fact atom-atom distance plot after an alignment.

We agree with the reviewer that the datapoints in the plots on **Figures 5D-G** do not represent the RMSD values, but instead the distances between the same CA atoms in the two aligned structures. We corrected the errors and introduced the correct labels into **Figures 5D-G** and **Figures S2-S3**.

The methods relating to how alignment between the two sets of atomic coordinates was performed and how the RMSD (or core RMSD) value was calculated were described briefly in the Methods section in the original draft. We expanded that paragraph to provide a more thorough explanation.

“2FoFc electron density map” should be “2Fo-Fc electron density map”. This may be trivial. It should be also mentioned if the map is weighted. If so, how it was weighted.

We agree with the reviewer’s suggestion. The typing error was corrected. The 2Fo-Fc maps are actually the sigmaA-weighted 2mFo-DFc maps generated in REFMAC or COOT. The Methods section was updated to clarify that. The 2Fo-Fc map was renamed as a “SigmaA- weighted 2Fo-Fc electron density map” throughout the text.

In the pMHC-TCR community, when a pMHC structure is presented, a common practice is to orient pMHC in such a way that the N-terminus of peptide is on the left and the C-terminus is on the right. Of course, authors certainly have their own ways to orient pMHC in a figure. But, in one figure, its orientation should be consistent. It is very confusing to orient one in one way and reverse the orientation in another, even for two parallel figures. Check Figure 4,6,7 and related supplementary figures.

We agree with the reviewer that, for convenience and especially for a general reader not familiar with details, maintaining the same or similar orientation of pMHC molecule in all the figures would be helpful in understanding the narrative. We have made corrections so that all the figures representing pMHC now have the uniform orientation. The only exception was made in **Figures S12D** and **S12E**, where the direction was 180° reversed for better viewing the molecular surface.

3. Problems in scientific conception. There are number of places in the text that are full of misconception and confusion. For example, on Page 21, the paragraph on TCR is very problematic.

“In every TCR, the complementarity determining regions (CDRs) responsible for epitope-MHC recognition are located at the extremities (CDR loops) of the variable (V α and V β) domains (Rudolph et al., 2006).”

What is “epitope-MHC” recognition?

The incorrect description of CDR was removed. The “epitope-MHC” recognition term was a typing error. The original intent was to outline the importance of epitope-TCR interactions, and to emphasize that CDR loops and certain epitope residues are among the main factors in determining TCR-pMHC recognition. The sentence was revised.

It sounds like the “CDRs” are CDR loops. The word “extremities” used here to describe CDR loops is incorrect and unprofessional. Do authors know the exact locations of CDRs in the V domain?

Again, we thank the reviewer for the thorough review of the manuscript. Using that term was incorrect and it was removed. This section was revised to conform with scientific terminology.

“The CDRs in TCR27 were defined based on similarity with other TCRs, ...”

Which other TCRs were used for the definition?

It is not a correct way for CDRs definition.

We agree that this point required clarification. To solve the crystal structure of TCR27 by molecular replacement, we have used the coordinates of a known TCR with the highest amino acid sequence identity to TCR27, which belongs to the TCR-pMHC structure PDB 1OGA. The RMSD value between TCR27 (TCR_1) and TCR 1OGA was 1.4Å and the AA sequence identity between their TCR α subunits was 93%. This was included in the text. The incorrect CDR definition was removed.

“The TCR27 structure is an $\alpha\beta$ heterodimer, where each monomer consists of one constant and one variable domain, all arranged into the constant and the variable regions, respectively”
Very confusing.

We agree with reviewer’s opinion, the above sentence was removed from the text.

“Unexpectedly, the quaternary conformation of TCR_1 was different from that of TCR_2, with an RMSD value between the coordinates of their C α atoms of 3.1Å.”

There is something totally wrong here. It is impossible to have such a huge RMSD value from an alignment of two conformations of one TCR. The authors need to understand the meaning of RMSD first. It is a term used in many places of the paper, such as in RMSD plots as commented above.

We agree that the sentence above was incorrectly written and should instead read “the quaternary conformation of TCR_1 was different from that of TCR_2, with an RMSD value between the two structures of 3.1Å (excluding CDR3 loops)”. The usage of terms such as “RMSD value” or “distance between CA atoms” was clarified throughout the text. The term RMSD was used when the two sets of atomic coordinates were aligned or superimposed and compared to each other to express the average distance between their identical atoms. The term “CA difference plot” was used in comparing the distances between each pair of identical C α atoms in the two aligned structures, and to identify the regions with largest deviations.

“The “hinge” areas around which the subunit “rotation” occurs (shown by the arrows in Fig.5D

and Fig.5G) is located at or close to the dimer interface between the two variable TCR domains.”

What is the meaning of “hinge areas”, which is located at or close the dimer interface?

We agree with the reviewer that the definition was unclear, and it was replaced with a correct term, which is center of rotation or rotation axis. The approximate center of one subunit rotation with respect to another one is located close to the AA residues with the smallest distance between the CA atoms of the two non-aligned subunits. The approximate position of each center of rotation is shown by the arrows in **Fig.5D and 5F**. To explain the mechanism of quaternary transition between TCR_1 and TCR_2 and to pinpoint the rotation axis, we aligned the individual domains (for instance, V α or V β as shown in Fig.5D and 5F) of TCR_1 and TCR_2. After that, we compared the distances between CA atoms of the *non-aligned domains*, from which the position of the approximate center of rotation was deduced. Such approach shows that the rigid body shift of one TCR subunit with respect to another one describes the most likely mechanism of transition between the two quaternary structures, TCR_1 and TCR_2.

4. Some potential technique issues.

Modeling issue. It is fine to make a prediction of a TCR-pMHC interaction model without their complex structure. But some general knowledges of the interaction between TCR and pMHC are required to judge the quality of a model. If in a model TCR interacts only to the peptide without important participation of MHC, the model is wrong.

We agree that the information provided regarding the pMHC-TCR27 complex was incomplete and therefore we supplemented the manuscript with additional data, added and corrected related figures. The coordinate file for the top docking solution was added to the SI section. A number of HLA residues (mainly in the helices 1 and 2) interacting with TCR27 (TCR “footprint”) were identified by NMR (**Fig.7D**) and used in the guided TCR-MHC docking. In the resultant TCR27-pMHC nearly all of these HLA residues were located at or close to the pMHC-TCR interface (**Fig.7E**). The data show that the contacts between TCR27 and pMHC within the complex involve TCR, HLA and bound epitope. To reflect that, we have revised the **Fig.7F**, which represents the TCR footprint to include all six CDR loops. In the SI section (**Fig. S8D**), we included a figure with a combined TCR footprint that shows all four top scoring docking solutions to illustrate the consistency and similarity between the docking results. The added **Fig.S9A,C,D** provides additional details on the pMHC-TCR interface.

Water molecules. All eleven crystal structures reported in this paper have resolution limits between 2.10 Å and 1.53 Å. Water molecules could be easily identified in these structures. It is well known that water molecule plays an important role in the formation of a hydrogen bond network between peptide and MHC. Strangely, authors never mentioned any water molecules in the analysis of these structures.

We agree that water molecules are an integral part of every protein crystal structure. Of special importance could be the waters inside the pockets, especially when shielded from the rest of the solvent by the ligand-protein interactions. For instance, we demonstrated that waters in the HLA-B7 pocket D of the peptide binding site may be substituted with small polar ligands such as glycerol, which in turn affect the bound epitope conformation (**Fig.S5**). However, the vast

Figure 1. Rfree vs resolution, from Shao et al., 2017.

majority of the MHC structures were isomorphous, including the same pattern for “buried” water molecules. Because of that, we decided to limit description of the water-mediated contacts to the most essential. To include more information of the role of water in these structures, we modified Fig.2C where included the buried water molecules in direct contacts with anchor residues (Arg-P3 and Ser-P6). Because of importance of residue P₄, we have included an additional Fig. S4 in the SI section to illustrate that the charged side chain at P₄ (phosphate or mimetic) is exposed to solvent, but not flexible and supports the hydrogen bond contacts with surrounding water molecules.

R factors in refinement. All eleven crystal structures in different resolution ranges have nearly the same R and Rfree factors, R=16.74(+/-)0.38% and Rfree=21.09(+/-)1.12%. Maybe, it is what truly happened. It is very rare.

We have re-checked the refinement data and they were correct except for 7RZJ (the typing error in Table S1 was corrected). We thoroughly validated the structure factors and the models using the PDB validation server, COOT and CCP4 tools (Sfcheck, Procheck, Molprobit and others). We re-checked the R/Rfree parameters and they remained close (within 0.5%) to the reported values. The majority (8 out of 11) of the crystal structures described in the manuscript were isomorphous and refined using the same method (REFMAC) which all contributed to similarity between their refinement parameters.

Introducing anisotropic refinement slightly improved R/Rfree for high-resolution structures, however, it also increased the bias in the ED maps and was not used.

The Rfree values for all the structures presented in this report also fall within the range of values expected for the reported high to medium-high resolution range (<1.6Å-2.1Å, figure 1 and the relevant references from RCSB team).

References:

Shao, C., Yang, H., Westbrook, J.D., Young, J.Y., Zardecki, C. and Burley, S.K., 2017. Multivariate analyses of quality metrics for crystal structures in the PDB archive. *Structure*, 25(3), pp.458-468.

Shao, C., Liu, Z., Yang, H., Wang, S. and Burley, S.K., 2018. Outlier analyses of the Protein Data Bank archive using a probability-density-ranking approach. *Scientific data*, 5(1), pp.1-11.

Reviewer #2 (Remarks to the Author):

The paper “Molecular mechanism of phosphopeptide neoantigen immunogenicity” is an excellent study of the characterization of an $\alpha\beta$ TCR interaction with phosphopeptide MHC complex in comparison to related pMHC. The authors thoroughly explore peptide binding space surrounding the target epitope using biophysical and functional techniques to obtain a convincing understanding of the binding events in question. There is good cross-fertilization and concordance between the various technologies that is supportive of the models presented. Overall, this is a well-designed study that is worthy of publication. There are some minor concerns that should be easily addressed.

1. The NMR assignments should be deposited in the Biological Magnetic Resonance database (BMRB) <https://bmrbl.io/>. If already deposited, then the accession number should be listed.

The NMR data have been deposited in the BMRB and the accession number 51815 ## is included in the manuscript.

2. The range of variation between HADDOCK docking overlays in Fig. S7 would be clearer if you could show a top down view of the highest scoring models like you show in Fig. 7G showing the differences in positioning of the six CDR loops or at least the CDR3s if that makes the figure clearer. Alternately it would be helpful to include a supplemental PDB file (not for deposition, but just for reader evaluation). As a reader, I would be very interested to have access to this model. The figure showing the residues used in the HADDOCK docking is very helpful but the residues should also be enumerated as simple text in the legend or a separate table.

We thank the reviewer for the suggestion. We have modified **Fig.7F** (TCR footprint) to show projection of the CDR loops over the MHC for the highest-scoring model. To compare the different docking solutions, we replaced **Fig.S8D** with a composite TCR footprint, similar to that in Figure 7F. The supplemental PDB file representing the top TCR-MHC docking solution is provided. The **Fig.S9A** shows the amino acid residues at the interface selected for guided docking.

3. It would be remiss not to mention the parallels your study has to a recent series of publications on the preTCR-pMHC interaction wherein X-ray structures were used in interpreting data from NMR to model a complex using HADDOCK (Mallis, et al J Biol Chem 2018, PMID29101227). The conclusions of that study were initially less direct due to the weak nature of the interaction, nevertheless, the follow-up studies did lead to actionable data and also to an X-ray structure (Mizsei, et al J Biol Chem 2021 PMID33837736 ; Li, et al Science 2021 PMID33335016). It is at your discretion whether to add mention of this in the discussion. We thank the reviewer for a very useful suggestion. Although these references pertain to interactions between pre-TCR and truncated MHC, we applied a similar approach to map the interactions between TCR and MHC. The authors (Mallis, et al J Biol Chem 2018, PMID29101227) used NMR mapping and Haddock docking to predict the tilting of pre-TCR, which was later corroborated by the X-ray structure (Li, et al Science 2021 PMID33335016).

These studies validated the combinatorial NMR/Haddock docking approach in acquiring the complex structure. We incorporated the above references in our manuscript, and added a related paragraph to the Discussion.

Reviewer #3 (Remarks to the Author):

The present manuscript “Molecular mechanism of phosphopeptide neoantigen immunogenicity” is a detailed and well laid-out study that nicely brings together biophysical, structural and cellular tools and approaches to gain additional mechanistic understanding of phosphopeptide neoantigen immunogenicity. The choice of orthogonal methods (e.g. NMR, Structure, Docking, Cellular assays) has been done very careful each adding valuable information to get a holistic mechanistic view on how phosphopeptide antigens are contributing to immunogenic response. A couple of minor suggestions to further improve the quality and clarity of the manuscript:

(1) In the Methods part you describe the use of biotinylated pMHC for the BLI studies. Please add how this protein material was prepared (in vivo vs. in vitro biotinylation?), including the level of biotinylation as assessed by MS.

We agree with the reviewer’s suggestion. We included the protocol for preparing the biotinylated proteins into the Methods section. The level of protein biotinylation in our samples approached 100%, as initially was assessed by MS. However, in this study instead of MS we routinely applied a gel-shift assay using streptavidin. To illustrate this assay, we included **Fig.S13** in the SI section of manuscript. This method helped to confirm protein biotinylation and quickly evaluate the stability of biotinylated proteins right before the experiment.

(2) In the Methods part you describe the E.coli medium used for the preparation of the triple labeled HLA-B*0702 protein. Please state the concentrations used for $^{15}\text{NH}_4\text{Cl}$ and $^{13}\text{C}, ^2\text{H}$ -glucose as well as the $^2\text{H}, ^{15}\text{N}, ^{13}\text{C}$ -celtone.

The corresponding numbers were included in the manuscript (Methods).

(3) BLI was employed to characterize the physical interactions between pMHC ligands and TCR27, which clearly showed that the binding affinity between pMHC and TCR27 is dependent on the nature of the amino acid residue at P4. For transparency it will be important to share the raw data of the sensorgrams in the main manuscript together with the associated equilibrium-binding data fits (plus kinetic fits for interactions that have clear kinetics features that can be analyzed). Particular the kinetics values can further our understanding of the molecular interactions that are taking place. Furthermore it will be important to add error bars (SEM) on Figure 2A as they are not visible in the plot but SEM values are given when presenting the affinity values.

We added a detailed description of the BLI protocol to the corresponding Methods section. Since it is more appropriate for illustrating BLI data to show the actual sensorgrams, than recalculated data, we replaced the **Fig.2A** with the sensorgrams and the curve fitting analysis (revised **Fig.2A** and added **Fig.S9B**). Sensorgrams do not have SEM values. However, in each run we included at least one concentration duplicate to provide a better error estimate, as recommended by the literature and by manufacturer. The differences between duplicates were

less than 3%. On and off-rates were too fast for calculation by this method and we have determined the dissociation constant values only. For measurements and data fitting/analysis, we used a BLI Octet Red96 instrument and related software, as previously reported (Gao et al, 2018; Congdon et al, 2022).

Gao, A., Vasilyev, N., Luciano, D.J., Levenson-Palmer, R., Richards, J., Marsiglia, W.M., Traaseth, N.J., Belasco, J.G. and Serganov, A., 2018. Structural and kinetic insights into stimulation of RppH-dependent RNA degradation by the metabolic enzyme DapF. *Nucleic acids research*, 46(13), pp.6841-6856.

Congdon, E.E., Pan, R., Jiang, Y., Sandusky-Beltran, L.A., Dodge, A., Lin, Y., Liu, M., Kuo, M.H., Kong, X.P. and Sigurdsson, E.M., 2022. Single domain antibodies targeting pathological tau protein: Influence of four IgG subclasses on efficacy and toxicity. *Ebiomedicine*, 84, p.104249.

REVIEWERS' COMMENTS

Reviewer #1 (Remarks to the Author):

The revised manuscript has been significantly improved. All of my concerns raised previously have been addressed.

Reviewer #2 (Remarks to the Author):

The authors have addressed all of my concerns satisfactorily.

Reviewer #3 (Remarks to the Author):

Dear authors,

thanks for addressing all the reviewers comments appropriately and in great detail. Again, I would like to congratulate you for the excellent work, the nice study outline and clarity of the results and conclusions, which I'm sure will attract a fair amount of attention.